# Lotus: Diffusion-based Visual Foundation Model for High-quality Dense Prediction

**Jing He[1]\* Haodong Li[1]\* Wei Yin[2] Yixun Liang[1] Leheng Li[1] Kaiqiang Zhou[3] Hongbo Zhang[3] Bingbing Liu[3] Yingcong Chen[1,4]** ✉

[1]HKUST(GZ) [2]University of Adelaide [3]Noah's Ark Lab [4]HKUST
{jhe812, hli736}@connect.hkust-gz.edu.cn; yingcongchen@ust.hk

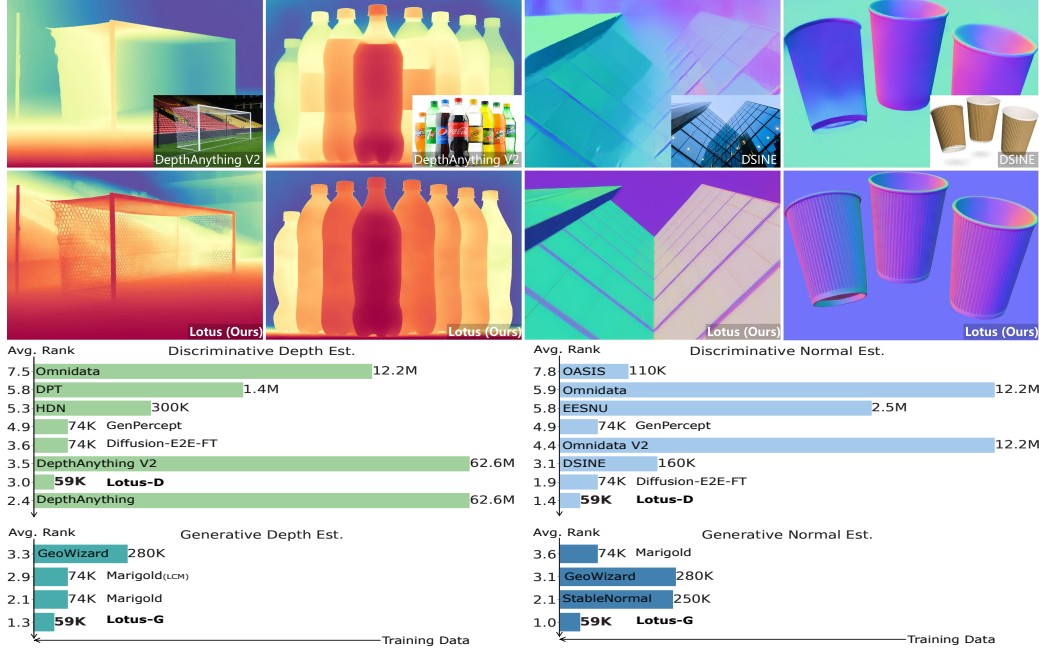

Figure 1: We present **Lotus**, a diffusion-based visual foundation model for dense geometry prediction. With minimal training data, Lotus achieves promising performance in zero-shot depth and normal estimation. "Avg. Rank" indicates the average ranking across all metrics, where lower values are better. Bar length represents the amount of training data used.

## Abstract

Leveraging the visual priors of pre-trained text-to-image diffusion models offers a promising solution to enhance zero-shot generalization in dense prediction tasks. However, existing methods often uncritically use the original diffusion formulation, which may not be optimal due to the fundamental differences between dense prediction and image generation. In this paper, we provide a systemic analysis of the diffusion formulation for the dense prediction, focusing on both quality and efficiency. And we find that the original parameterization type for image generation, which learns to predict noise, is harmful for dense prediction; the multi-step noising/denoising diffusion process is also unnecessary and challenging to optimize. Based on these insights, we introduce **Lotus**, a diffusion-based visual foundation model with a simple yet effective adaptation protocol for dense prediction. Specifically, Lotus is trained to directly predict annotations instead of noise, thereby avoiding harmful variance. We also reformulate the diffusion process into a single-step procedure, simplifying optimization and significantly boosting inference speed. Additionally, we introduce a novel tuning strategy called detail preserver, which achieves more accurate and fine-grained predictions. Without scaling up the training data or model capacity, Lotus achieves promising performance in zero-shot depth and normal estimation across various datasets. It also enhances efficiency, being significantly faster than most existing diffusion-based methods. Lotus' superior quality and efficiency enables a wide range of practical applications, such as joint estimation, single/multi-view 3D reconstruction, etc.

## 1 INTRODUCTION

Dense prediction is a fundamental task in computer vision, benefiting a wide range of applications, such as 3D/4D reconstruction (Huang et al., 2024; Long et al., 2024; Wang et al., 2024; Lei et al., 2024), tracking (Xiao et al., 2024; Song et al., 2024), and autonomous driving (Yurtsever et al., 2020; Hu et al., 2023). Estimating pixel-level geometric attributes from a single image requires comprehensive scene understanding. Although deep learning has advanced dense prediction, progress is limited by the quality, diversity, and scale of training data, leading to poor zero-shot generalization. Instead of merely scaling data and model size, recent works (Lee et al., 2024; Ke et al., 2024; Fu et al., 2024; Xu et al., 2024) leverage diffusion priors for zero-shot dense prediction. These studies demonstrate that text-to-image diffusion models like Stable Diffusion (Rombach et al., 2022), pretrained on billions of images, possess powerful and comprehensive visual priors to elevate dense prediction performance. However, most of these methods directly inherit the pre-trained diffusion models for dense prediction tasks, without exploring more suitable diffusion formulations. This oversight often leads to challenging issues. For example, Marigold (Ke et al., 2024) directly fine-tunes Stable Diffusion for image-conditioned depth generation. While it significantly improves depth estimation, its performance is still constrained by overlooking the fundamental differences between dense prediction and image generation. Especially, its efficiency is also severely limited by standard iterative denoising processes and ensemble inferences.

Motivated by these concerns, we systematically analyze the diffusion formulation, trying to find a better formulation to fit the pre-trained diffusion model into dense prediction. Our analysis yields several important findings: ① The widely used parameterization, *i.e.*, noise prediction, for diffusion-based image generation is ill-suited for dense prediction. It results in large prediction errors due to harmful prediction variance at initial denoising steps, which are subsequently propagated and magnified throughout the entire denoising process (Sec. 4.1). ② Multi-step diffusion formulation is computation-intensive and is prone to sub-optimal with limited data and resources. These factors significantly hinder the adaptation of diffusion priors to dense prediction tasks, leading to decreased accuracy and efficiency (Sec. 4.2). ③ Though remarkable performance achieved, we observed that the model usually outputs vague predictions in highly-detailed areas (Fig. 8). This vagueness is attributed to catastrophic forgetting: the pre-trained diffusion models gradually lose their ability to generate detailed regions during fine-tuning (Sec. 4.3).

Following our analysis, we propose **Lotus**, a diffusion-based visual foundation model for dense prediction, featuring a simple yet effective fine-tuning protocol (see Fig. 2). First, Lotus is trained to directly predict annotations, thereby avoiding the harmful variance associated with standard noise prediction. Next, we introduce a one-step formulation, *i.e.*, one step between pure noise and clean output, to facilitate model convergence and achieve better optimization performance with limited high-quality data. It also considerably boosts both training and inference efficiency. Moreover, we implement a novel detail preserver through a task switcher, allowing the model either to generate annotations or reconstruct the input images. It can better preserve the fine-grained details in the input image during dense annotation generation, achieving

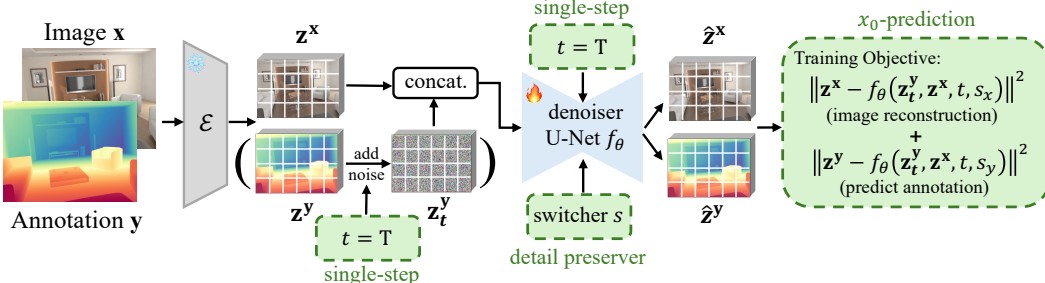

Figure 2: **Adaptation protocol of Lotus.** After the pre-trained VAE encoder $\mathcal{E}$ encodes the image **x** and annotation **y** to the latent space: ① the denoiser U-Net model $f_\theta$ is fine-tuned using $x_0$-prediction; ② we employ single-step diffusion formulation at time-step $t = T$ for better convergence; ③ we propose a novel detail preserver, to switch the model either to reconstruct the image or generate the dense prediction via a switcher $s$, ensuring a more fine-grained prediction. The noise $\mathbf{z_T^y}$ in bracket is used for our generative **Lotus-G** and is omitted for the discriminative **Lotus-D**.

higher performance without compromising efficiency, requiring additional parameters, or being affected by surface textures.

To validate Lotus, we conduct extensive experiments on two primary geometric dense prediction tasks: zero-shot monocular depth and normal estimation. The results demonstrate that Lotus achieves promising, and even superior, performance on these tasks across a wide range of evaluation datasets. Compared to traditional discriminative methods, Lotus delivers remarkable results with only 59K training samples. Among generative approaches, Lotus also outperforms previous methods in both accuracy and efficiency, being significantly faster than methods like Marigold (Ke et al., 2024) (Fig. 3). Beyond these improvements, Lotus seamlessly supports various applications, *e.g.*, joint estimation, single/multi-view 3D reconstruction, *etc*.

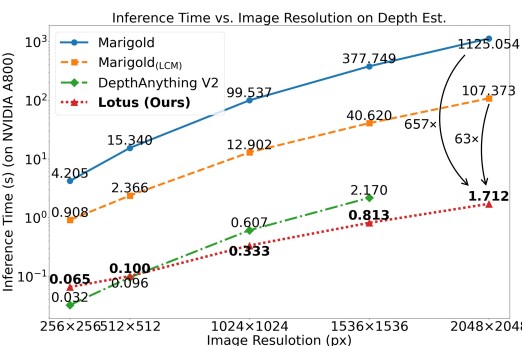

Figure 3: **Inference time comparison in depth estimation between Lotus and SoTA methods.** Lotus is hundreds of times faster than Marigold and slightly faster than DepthAnything V2 at high resolutions. DepthAnything V2's inference time at $2048 \times 2048$ is not plotted because it requires $> 80$GB graphic memory.

In conclusion, our key contributions are as follows:

- We systematically analyze the diffusion formulation and find their parameterization type, designed for image generation, is unsuitable for dense prediction and the computation-intensive multi-step diffusion process is also unnecessary and challenging to optimize.
- We propose a novel detail preserver that ensures more accurate dense predictions especially in detail-rich areas, without compromising efficiency, introducing additional network parameters, or being affected by surface textures.
- Based on our insights, we introduce **Lotus**, a diffusion-based visual foundation model for dense prediction with simple yet effective fine-tuning protocol. Lotus achieves promising performance on both zero-shot monocular depth and surface normal estimation. It also enables a wide range of applications.

## 2 RELATED WORKS

### 2.1 TEXT-TO-IMAGE GENERATIVE MODELS

In the field of text-to-image generation, the evolution of methodologies has transitioned from generative adversarial networks (GANs) (Goodfellow et al., 2014; Zhang et al., 2017; 2018; 2021; He et al., 2022; Karras et al., 2019; 2020; 2021; Zhang et al., 2017; 2018; Xu et al., 2018; Zhang et al., 2021) to advanced diffusion models (Ho et al., 2020; Ramesh et al., 2022; Saharia et al., 2022; Ramesh et al., 2021; Nichol et al., 2021; Chen et al., 2023; Rombach et al., 2022; Ramesh et al., 2021). A series of diffusion-based methods such as GLIDE (Nichol et al., 2021), DALL·E2 (Ramesh et al., 2022), and Imagen (Saharia et al., 2022) have been introduced, offering enhanced image quality and textual coherence. The Stable Diffusion (SD) (Rombach et al., 2022), trained on large-scale LAION-5B dataset (Schuhmann et al., 2022), further enhances the generative quality, becoming the community standard. In our paper, we aim to leverage the comprehensive and encyclopedic visual priors of SD to facilitate zero-shot generalization for dense prediction tasks.

### 2.2 GENERATIVE MODELS FOR DENSE PERCEPTION

Currently, a notable trend involves adopting pre-trained generative models, particularly diffusion models, into dense prediction tasks. Marigold (Ke et al., 2024) and GeoWizard (Fu et al., 2024) directly apply the standard diffusion formulation and the pre-trained parameters, without addressing the inherent differences between image generation and dense prediction, leading to constrained performance. Their efficiency is also severely limited by standard iterative denoising processes and ensemble inferences. In this paper, we propose a novel diffusion formulation tailored to the dense prediction. Aiming to fully leveraging the pre-trained diffusion's powerful visual priors, Lotus enables more accurate and efficient predictions, finally achieving promising performance.

More recent works, GenPercept (Xu et al., 2024) and StableNormal (Ye et al., 2024), also adopted single-step diffusion. However, GenPercept (Xu et al., 2024) first removes noise input for deter-

ministic characteristic based on DMP (Lee et al., 2024), and then adopts one-step strategy to avoid surface texture interference. It lacks systematic analysis of the diffusion formulation, only treats the U-Net as a deterministic backbone and still falls short in performance. In contrast, Lotus systematically analyzes the standard stochastic diffusion formulation for dense prediction and proposes innovations such as the detail preserver to improve accuracy especially in detailed area, finally delivering much better results (Tab. 1). Additionally, Lotus is a stochastic model. In contrast to GenPercept's deterministic nature, Lotus enables uncertainty predictions. StableNormal (Ye et al., 2024) predicts normal maps through a two-stage process. While the first stage produces coarse normal maps with single-step diffusion, the second stage performs refinement still with iterative diffusion which is computation-intensive. In comparison, Lotus not only achieves fine-grained predictions thanks to our novel detail preserver without extra stages or parameters, but also delivers much superior results (Tab. 2) thanks to our designed diffusion formulation that better fits the pre-trained diffusion for dense prediction. Recently, a concurrent work, Diffusion-E2E-FT (Garcia et al., 2024), has also achieved promising results in a single step. Its main contribution lies in addressing the issue where Marigold (Ke et al., 2024) and similar models (Fu et al., 2024) use inconsistent pairings of time-step and noise, resulting in poor predictions. By setting the "time-step spacing" to "trailing" mode in schedulers, it prevents "GT" signal leakage during inference, improving accuracy. While the performance of Lotus-D and Diffusion-E2E-FT is similar, Lotus is based on a systematic analysis of stochastic diffusion for dense prediction, with innovations like the detail preserver to enhance accuracy, particularly in detailed areas. Additionally, unlike the deterministic Diffusion-E2E-FT, Lotus (Lotus-G) is a stochastic model that enables uncertainty predictions.

## 2.3 Monocular Depth and Normal Prediction

Monocular depth and normal prediction are two crucial dense prediction tasks. Solving them typically demands comprehensive scene understanding capability. Starting from (Eigen et al., 2014), early CNN-based methods for depth prediction, such as (Fu et al., 2018), (Lee et al., 2019), (Yuan et al., 2022), focus only on specific domains. Subsequently, in pursuit of a generalizable depth estimator, many methods expand model capacity and train on larger and more diverse datasets, such as DiverseDepth (Yin et al., 2021a) and MiDaS (Ranftl et al., 2020). DPT (Ranftl et al., 2021) and Omnidata (Eftekhar et al., 2021) are further proposed based on vision transformer (Ranftl et al., 2021), significantly enhancing performance. LeRes (Yin et al., 2021b) and HDN (Zhang et al., 2022) further introduce novel training strategies and multi-scale depth normalization to improve predictions in detailed areas. More recently, the DepthAnything series (Yang et al., 2024a;b) and Metric3D series (Yin et al., 2023; Hu et al., 2024) collect and leverage millions of training data to develop more powerful estimators. Normal prediction follows the same trend. Starting with the early CNN-based methods like OASIS (Chen et al., 2020), EESNU (Bae & Davison, 2021) and Omnidata series (Eftekhar et al., 2021; Kar et al., 2022) expand the model capacity and scale up the training data. Recently, DSINE (Bae & Davison, 2024) achieves SoTA performance by rethinking inductive biases for surface normal estimation. In our paper, we focus on leveraging pre-trained diffusion priors to enhance zero-shot dense predictions, rather than expanding model capacity or relying on large training data, which avoids the need for intensive resources and computation.

## 3 Preliminaries

**Diffusion Formulation for Dense Prediction.** Following Ke et al. (2024) and Fu et al. (2024), we also formulate dense prediction as an image-conditioned annotation generation task based on Stable Diffusion (Rombach et al., 2022), which performs the diffusion process in low-dimensional latent space for computational efficiency. First, the auto-encoder, which consists an encoder $\mathcal{E}(\cdot)$ and a decoder $\mathcal{D}(\cdot)$, is trained to map between RGB space and latent space, *i.e.*, $\mathcal{E}(\mathbf{x}) = \mathbf{z}^{\mathbf{x}}$, $\mathcal{D}(\mathbf{z}^{\mathbf{x}}) \approx \mathbf{x}$. The auto-encoder also maps between dense annotations and latent space effectively, *i.e.*, $\mathcal{E}(\mathbf{y}) = \mathbf{z}^{\mathbf{y}}$, $\mathcal{D}(\mathbf{z}^{\mathbf{y}}) \approx \mathbf{y}$ (Ke et al., 2024; Fu et al., 2024; Xu et al., 2024; Ye et al., 2024). Following Ho et al. (2020), Stable Diffusion establishes a pair of *forward* nosing and *reversal* denoising processes in latent space. In *forward* process, Gaussian noise is gradually added at levels $t \in [1, T]$ into sample $\mathbf{z}^{\mathbf{y}}$ to obtain the noisy sample $\mathbf{z_t^y}$:

$$\mathbf{z_t^y} = \sqrt{\overline{\alpha}_t}\mathbf{z^y} + \sqrt{1 - \overline{\alpha}_t}\epsilon, \tag{1}$$

where $\epsilon \sim \mathcal{N}(0, I)$, $\overline{\alpha}_t := \prod_{s=1}^{t}(1 - \beta_s)$, and $\{\beta_1, \beta_2, \ldots, \beta_T\}$ is the noise schedule with $T$ steps. At time-step $T$, the sample $\mathbf{z^y}$ is degraded to pure Gaussian noise. In the *reversal* process, a neural network $f_\theta$, usually a U-Net model (Ronneberger et al., 2015), is trained to iteratively remove noise from $\mathbf{z_t^y}$ to predict the clean sample $\mathbf{z^y}$. The network is trained by sampling a random $t \in [1, T]$ and minimizing the loss function $L_t$.

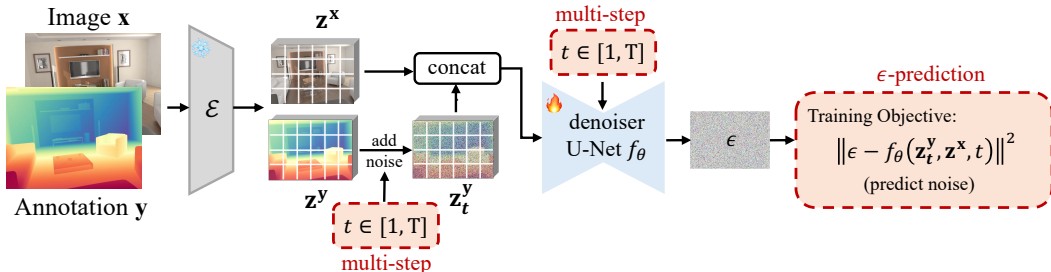

Figure 4: **Adaptation protocol of Direct Adaptation.** Starting with a pre-trained Stable Diffusion model, image **x** and annotation **y** are encoded using the pre-trained VAE. The noisy annotation $\mathbf{z_t^y}$ is obtained by adding noise at level $t \in [1, T]$. The U-Net input layer is coupled to accommodate the concatenated inputs and then fine-tuned using the standard diffusion objective, $\epsilon$-prediction, under the original multi-step formulation.

**Parameterization Types.** To enable gradient computation for network training, there are two basic parameterizations of the loss function $L_t$. ① $\epsilon$-prediction (Ho et al., 2020): the model $f_\theta$ learns to predict the added noise $\epsilon$; ② $x_0$-prediction (Ho et al., 2020): the model $f_\theta$ learns to directly predict the clean sample $\mathbf{z^y}$. The loss functions for these parameterizations are formulated as:

$$\epsilon\text{-prediction: } L_t^\epsilon = ||\epsilon - f_\theta^\epsilon(\mathbf{z_t^y}, \mathbf{z^x}, t)||^2,$$
$$x_0\text{-prediction: } L_t^{\mathbf{z}} = ||\mathbf{z^y} - f_\theta^{\mathbf{z}}(\mathbf{z_t^y}, \mathbf{z^x}, t)||^2. \tag{2}$$

where $f_\theta^*$ is the denoiser model to be learnt, $* \in \{\epsilon, \mathbf{z}\}$. $\epsilon$-prediction is commonly chosen as the standard for parameterizing the denoising model, as it empirically achieves high-quality image generation with fine details and realism.

**Denoising Process.** DDIM (Song et al., 2020) is a key technique for multi-step diffusion models to achieve fast sampling, which implements an implicit probabilistic model that can significantly reduce the number of denoising steps while maintaining output quality. Formally, the denoising process from $\mathbf{z_\tau^y}$ to $\mathbf{z_{\tau-1}^y}$ is:

$$\mathbf{z_{\tau-1}^y} = \sqrt{\overline{\alpha}_{\tau-1}}\hat{\mathbf{z}}_\tau^{\mathbf{y}} + \text{direction}(\mathbf{z_\tau^y}) + \sigma_\tau\epsilon_\tau, \tag{3}$$

where $\hat{\mathbf{z}}_\tau^{\mathbf{y}}$ is the predicted clean sample at the denoising step $\tau$, direction($\mathbf{z_\tau^y}$) represents the direction pointing to $\mathbf{z_\tau^y}$ and $\sigma_\tau$ can be set to 0 if deterministic denoising is needed. And $\tau \in \{\tau_1, \tau_2, \dots, \tau_S\}$, an increasing sub-sequence of the time-step set $[1, T]$, is used for fast sampling. During inference, DDIM iteratively denoises the sample from $\tau_S$ to $\tau_1$ to obtain the clean one.

## 4 METHODOLOGY

We start our analysis by directly adapting the original diffusion formulation with minimal modifications as illustrated in Fig. 4. We call this starting point as "*Direct Adaptation*"[1]. Direct Adaptation is optimized using the standard diffusion objective as formulated in Eq. 2 (first row) and inferred by standard multi-step DDIM sampler. As shown in Tab. 3, Direct Adaptation fails to achieve satisfactory performance. In following sections, we will systematically analyze the key factors that affect adaptation performance step by step: parameterization types (Sec. 4.1); number of time-steps (Sec. 4.2); and the novel detail preserver (Sec. 4.3).

### 4.1 PARAMETERIZATION TYPES

The type of parameterization is crucial, it not only determines the loss function discussed in Sec. 3, but also influences the inference process (Eq. 3). During inference, the predicted clean sample $\hat{\mathbf{z}}_\tau^{\mathbf{y}}$, a key component in Eq. 3, is calculated according to different parameterizations [2].

$$\epsilon\text{-prediction: } \hat{\mathbf{z}}_\tau^{\mathbf{y}} = \frac{1}{\sqrt{\overline{\alpha}_\tau}}(\mathbf{z_\tau^y} - \sqrt{1 - \overline{\alpha}_\tau}f_\theta^\epsilon(\mathbf{z_\tau^y}, \mathbf{z^x}, \tau)),$$
$$x_0\text{-prediction: } \hat{\mathbf{z}}_\tau^{\mathbf{y}} = f_\theta^{\mathbf{z}}(\mathbf{z_\tau^y}, \mathbf{z^x}, \tau). \tag{4}$$

In the community, $\epsilon$-prediction is chosen as the standard for image generation. However, it is not effective for dense prediction task. In the following, we will discuss the impact of different parameterization types in denoising inference process for dense prediction task.

---

[1]Details of Direct Adaptation will be provided in the supplementary materials.

[2]The latest parameterization, $v$-prediction, combines $\epsilon$-prediction and $x_0$-prediction, producing results that are intermediate between the two. Please see the supplementary materials for more details.

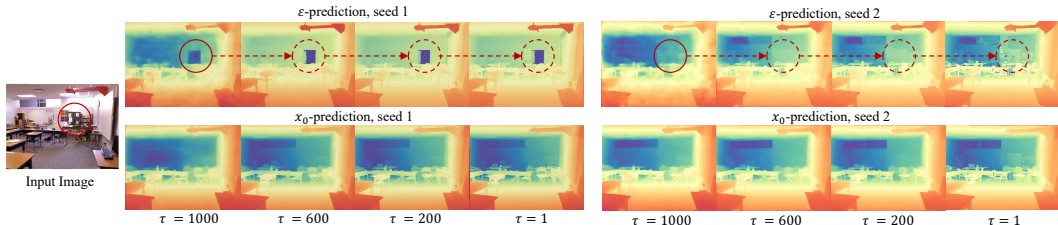

Figure 5: **Comparisons among different parameterizations using various seeds.** All models are trained on *Hypersim* (Roberts et al., 2021) and tested on the input image for depth estimation. The standard DDIM sampler is used with 50 denoising steps. Four steps are selected for clear illustration. From left (larger $\tau$) to right (smaller $\tau$) is the iterative denoising process.

Insights from the literature (Benny & Wolf, 2022; Salimans & Ho, 2022) reveal that $\epsilon$-prediction introduces larger pixel variance compared to $x_0$-prediction, especially at the initial denoising steps (large $\tau$). This variance mainly originates from the noise input. Specifically, for $\epsilon$-prediction in Eq. 4, at initial denoising step, $\tau \to T$, the value $\frac{1}{\sqrt{\bar{\alpha}_\tau}} \to +\infty$. Thus, the prediction variance from $f_\theta^\epsilon(\mathbf{z}_\tau^\mathbf{y}, \mathbf{z}^\mathbf{x}, \tau)$ will be amplified significantly, resulting in large variance of predicted $\hat{\mathbf{z}}_\tau^\mathbf{y}$. In contrast, there is no coefficient for $x_0$-prediction to re-scale the model output, achieving more stable predictions of $\hat{\mathbf{z}}_\tau^\mathbf{y}$ at initial denoising steps. Subsequently, the predicted $\hat{\mathbf{z}}_\tau^\mathbf{y}$ is used in Eq. 3, where its coefficient $\sqrt{\bar{\alpha}_{\tau-1}}$ are same across the two parameterizations, and other terms are of the same order of magnitude. Therefore, the $\hat{\mathbf{z}}_\tau^\mathbf{y}$ predicted by $\epsilon$-prediction, which has larger variance, exerts a more significant influence on denoising process. Since the process is iterative, this influence is continually preserved and maybe amplified.

We take the depth estimation as an example. During the inference process, we compute the predicted depth map $\hat{\mathbf{z}}_\tau^\mathbf{y}$ at each denoising step $\tau$. As illustrated in Fig. 5, the depth maps predicted by $\epsilon$-prediction significantly vary under different seeds while those predicted by $x_0$-prediction are more consistent. Although the large variance enhances diversity for image generation, it lead to unstable predictions in dense prediction tasks, potentially resulting in significant errors. For example in Fig. 5, the "dark gray cabinet" (highlighted in red circles) maybe wrongly considered as an "opened door" with significantly larger depth. While the predicted depth map *looks* more and more *plausible*, the error gradually propagates to the final prediction ($\tau = 1$) along the denoising process, indicating the persistent influence of the large variance. We further quantitatively measure the predicted depth maps by the absolute mean relative error (AbsRel) on NYUv2 dataset (Silberman et al., 2012). As shown in Fig. 6, $\epsilon$-prediction exhibits higher error with

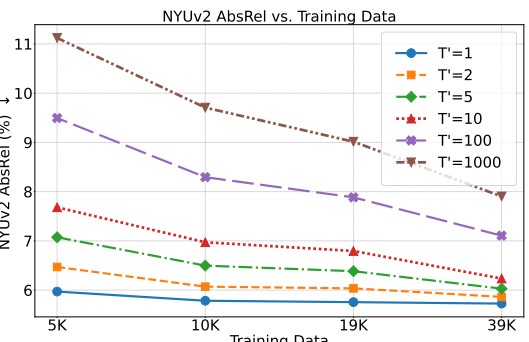

Figure 6: **Quantitative evaluation of the predicted depth maps $\hat{\mathbf{z}}_\tau^\mathbf{y}$ along the denoising process.** The experimental settings are same as Fig. 5. Six steps are selected for illustration. The banded regions around each line indicate the variance, wider areas representing larger variance.

Figure 7: **Comparisons among various training time-steps and data scales** evaluated on NYUv2 in depth estimation. All models are fine-tuned on *Hypersim* using $x_0$-prediction. During inference, if $T' > 50$, the DDIM sampler is used with 50 denoising steps; otherwise, the number of denoising steps is equal to $T'$. The results demonstrate improved performance with decreased training time-steps. The single-step diffusion formulation ($T' = 1$) exhibits best performance across different data volumes.

much larger variance compared to $x_0$-prediction at the initial denoising steps ($\tau \to T$), and the prediction error propagates with a higher slope. In contrast, $x_0$-prediction, directly predicting $\hat{\mathbf{z}}_\tau^\mathbf{y}$ without any coefficients to amplify the prediction variance, yields more stable and correct dense predictions than $\epsilon$-prediction. In conclusion, to mitigate the errors from large variance that ad-

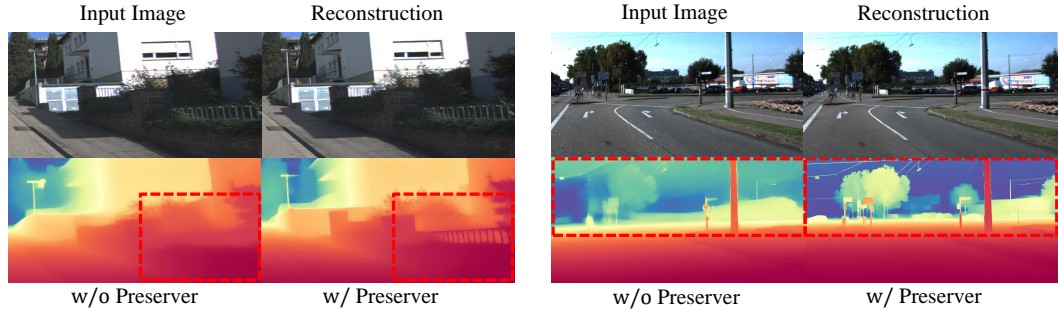

Figure 8: **Depth maps $w/$ and $w/o$ the detail preserver and reconstruction outputs.** Fine-tuning the diffusion model for dense prediction tasks can potentially degrade its ability to generate highly detailed images, resulting in blurred predictions in regions with rich detail. To preserve these fine-grained details, we introduce a detail preserver that incorporates an additional reconstruction task, enhancing the model's capacity to produce more accurate dense annotations.

versely affect the performance of dense prediction, we replace the standard $\epsilon$-prediction with the more tailored $x_0$-prediction.

## 4.2 NUMBER OF TIME-STEPS

Although $x_0$-prediction can improve the prediction quality, the multi-step diffusion formulation still leads to the propagation of predicted errors during the denoising process (Fig. 5, 6). Furthermore, utilizing multiple time-steps enhances the model's capacity, typically requiring large-scale training data to optimize and is beneficial for complex tasks such as image generation. However, for simpler tasks like dense prediction, where large-scale, high-quality training data is also scarce, employing multiple time-steps can make the model difficult to optimize. Additionally, training/inferring a multi-step diffusion model is slow and computation-intensive, hindering its practical application.

Therefore, to address these challenges, we propose fine-tuning the pre-trained diffusion model with fewer training time steps. Specifically, the original set of training time-steps is defined as $[1, T] = \{1, 2, 3, \ldots, T\}$, where $T$ denotes the total number of original training time-steps. We fine-tune the pre-trained diffusion model using a sub-sequence derived from this set. We define the length of this sub-sequence as $T'$, where $T' \leqslant T$ and $T$ is divisible by $T'$. This sub-sequence is obtained by evenly sampling the original set at intervals, defined as:

$$\{t_i = i \cdot k \mid i = 1, 2, \ldots, T'\}, \tag{5}$$

where $k = T/T'$ is the sampling interval. During inference, the DDIM denoises the sample from noise to annotation using the same sub-sequence if $T' \leqslant 50$, otherwise we use 50 denoising steps.

As illustrated in Fig. 7, we conduct experiments by varying the number of time-steps $T'$ under $x_0$-prediction. The results clearly show that the performance gradually improves as the number of time-steps is reduced, no matter the training data scales, culminating in the best result when reduced to only a single step. We further consider more strict scenarios with more limited training data to assess its impact on model optimization. As depicted in Fig. 7, these experiments reveal that the multi-step formulation is more sensitive to increases in training data scales compared with single-step. Notably, the single-step formulation consistently yields lower prediction errors and demonstrates greater stability. Although it is conceivable that multi-step and single-step formulations might achieve comparable performance with unlimited high-quality data, it's expensive and sometimes impractical in dense prediction.

Decreasing the number of denoising steps can reduce the optimization space of the diffusion model, leading to more effective and efficient adaption, as suggested by the above phenomenon. Therefore, for better adaptation performance under limited resource, we reduce the number of training time-steps of diffusion formulation to only one, and fixing the only time-step $t$ to $T$. Additionally, the single-step formulation is much more computationally efficient. It also naturally prevents the harmful error propagation as discussed in Sec. 4.1, further enhancing the diffusion's adaptation performance in dense prediction.

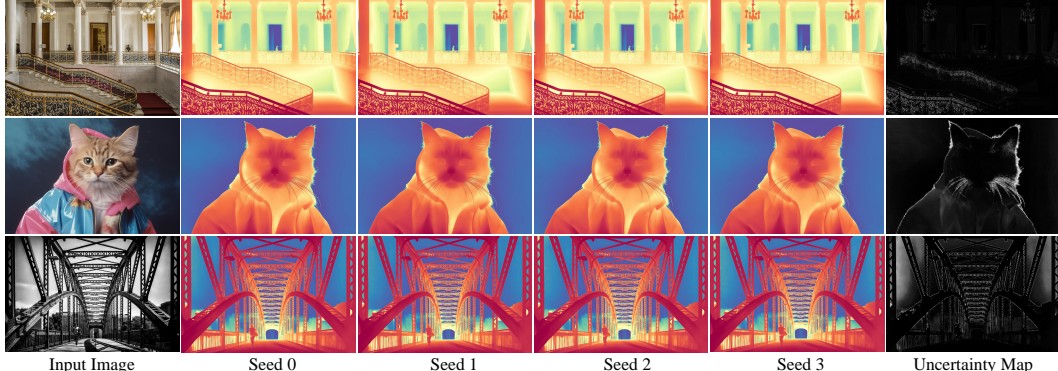

Figure 9: **Depth maps of multiple inferences and uncertainty maps.** Areas like the sky, object edges, and intricate details (*e.g.*, cat whiskers) typically exhibit high uncertainty.

### 4.3 DETAIL PRESERVER

Despite the effectiveness of the above designs, the model still struggles with processing detailed areas (Fig. 8, $w/o$ Preserver). The original diffusion model excels at generating detailed images. However, when adapted to predict dense annotations, it can lose such detailed generation ability, due to unexpected catastrophic forgetting (Zhai et al., 2023; Du et al., 2024). This leads to challenges in predicting dense annotations in intricate regions.

To preserve the rich details of the input images, we introduce a novel regularization strategy called *Detail Preserver*. Inspired by previous works (Long et al., 2024; Fu et al., 2024), we utilize a task switcher $s \in \{s_x, s_y\}$, enabling the denoiser model $f_\theta$ to either generate annotation or reconstruct the input image. When activated by $s_y$, the model focuses on predicting annotation. Conversely, when $s_x$ is selected, it reconstructs the input image. The switcher $s$ is a one-dimensional vector encoded by the positional encoder and then added with the time embeddings of diffusion model, ensuring seamless domain switching without mutual interference. This dual capability enables the diffusion model to make detailed predictions and thus leading to better performance. Overall, the loss function $L_t$ is:

$$L_t = ||\mathbf{z^x} - f_\theta(\mathbf{z_t^y}, \mathbf{z^x}, t, s_x)||^2 + ||\mathbf{z^y} - f_\theta(\mathbf{z_t^y}, \mathbf{z^x}, t, s_y)||^2, \tag{6}$$

where $t = T$ and thus $\mathbf{z_t^y}$ is a pure Gaussian noise.

### 4.4 STOCHASTIC NATURE OF DIFFUSION MODEL

One major characteristic of generative models is their stochastic nature, which, in image generation, enables the production of diverse outputs. In perception tasks like dense prediction, this stochasticity has the potential to allow the model generating predictions with uncertainty maps. Specifically, for any input image, we can conduct multiple inferences using different initialization noises and aggregate these predictions to calculate its uncertainty map. Thanks to our systematic analysis and tailored fine-tuning protocol, our method effectively reduces excessive flickering (large variance), only allowing for more

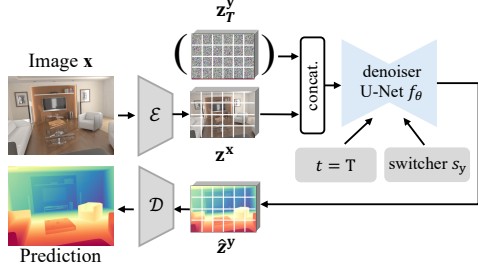

Figure 10: **Inference Pipeline of Lotus.** The noise $\mathbf{z_T^y}$ in bracket is used for **Lotus-G** and omitted for **Lotus-D**.

accurate uncertainty calculations in naturally uncertain areas, such as the sky, object edges, and fine details (*e.g.* cat whiskers), as shown in Fig. 9.

Most existing perception models are deterministic. To align with these, we can remove the noise input $\mathbf{z_t^y}$ and only input the encoded image features $\mathbf{z^x}$ to the U-Net denoiser. The model still performs well. In this paper, we finally present two versions of Lotus: **Lotus-G** (generative) with noise input and **Lotus-D** (discriminative) without noise input, catering to different needs.

## 4.5 INFERENCE

The inference pipeline is illustrated in Fig. 10. We initialize the annotation map with standard Gaussian noise $\mathbf{z_T^y}$, and encode the input image into its latent code $\mathbf{z^x}$. The noise $\mathbf{z_T^y}$ and the image $\mathbf{z^x}$ are concatenated and fed into the denoiser U-Net model. In our single-step formulation, we set $t = T$ and the switcher to $s_y$. The denoiser U-Net model then predicts the latent code of the annotation map. The final annotation map is decoded from the predicted latent code via the VAE decoder. For deterministic prediction, we eliminate the Gaussian noise $\mathbf{z_T^y}$ and only feed the latent code of the input image into U-Net.

## 5 EXPERIMENTS

### 5.1 EXPERIMENTAL SETTINGS

**Implementation details.** We implement Lotus based on Stable Diffusion V2 (Rombach et al., 2022), without text conditioning. During training, we fix the time-step $t = 1000$. For depth estimation, we predict in disparity space, *i.e.*, $d = 1/d'$, where $d$ represents the values in disparity space and $d'$ denotes the true depth. For more details, please see the supplementary materials.

**Training Datasets.** Both depth and normal estimation are trained on two synthetic dataset covering indoor and outdoor scenes: ① *Hypersim* (Roberts et al., 2021) is a photorealistic synthetic dataset featuring 461 indoor scenes. We use the official training split, which contains approximately 54K samples. After filtering out incomplete samples, around 39K samples remain, all resized to $576 \times 768$ for training. ② *Virtual KITTI* (Cabon et al., 2020) is a synthetic street-scene dataset with five urban scenes under various imaging and weather conditions. We utilize four of these scenes for training, comprising about 20K samples. All samples are cropped to $352 \times 1216$, with the far plane at 80m.

Following Marigold (Ke et al., 2024), we probabilistically choose one of the two datasets and then draw samples from it for each batch (*Hypersim* 90% and *Virtual KITTI* 10%).

**Evaluation Datasets and Metrics.** ① For zero-shot affine-invariant depth estimation, we evaluate Lotus on NYUv2 (Silberman et al., 2012), ScanNet (Dai et al., 2017), KITTI (Geiger et al., 2013), ETH3D (Schops et al., 2017), and DIODE (Vasiljevic et al., 2019) using absolute mean relative error (*AbsRel*), and also report $\delta 1$ and $\delta 2$ values. ② For surface normal prediction, we employ NYUv2, ScanNet, iBims-1 (Koch et al., 2018), Sintel (Butler et al., 2012) and OASIS (Chen et al., 2020) datasets, reporting mean angular error (*m.*) as well as the percentage of pixels with an angular error below $11.25°$ and $30°$. Please see supplementary materials for further details on the evaluation datasets and metrics.

### 5.2 QUANTITATIVE COMPARISONS

① For depth estimation (Tab. 1), Lotus-G demonstrates promising performance across all evaluation datasets, achieving the overall best rank compared to other generative baselines. Notice that we only require single step denoising process, significantly boosting the inference speed as shown in Fig. 3. Lotus-D also performs well, achieving comparable results to DepthAnything series. It is worthy to notice that Lotus is trained on only 0.059M images compared to DepthAnything's 62.6M images. ② For normal estimation (Tab. 2), both Lotus-G and Lotus-D outperform all other generative and discriminative methods in terms of average ranking. Please see the supplementary materials for **Qualitative Comparisons**.

### 5.3 ABLATION STUDY

As shown in Tab. 3, we conduct ablation studies to validate our designs. Starting with "Direct Adaptation", we incrementally test the effects of different components, such as parameterization types, the single-step diffusion process, and the detail preserver. Initially, we train the model using only the *Hypersim* dataset to establish a baseline. We then expand the training dataset using a mixture dataset strategy by including *Virtual KITTI*, aiming to enhance the model's generalization ability across different domains. For depth estimation, we further train the model in the disparity space to improve the accuracy. The findings from these ablations validate the effectiveness of our proposed adaptation protocol, demonstrating that each design plays a vital role in optimizing the diffusion models for dense prediction tasks.

# 6 CONCLUSION

In this paper, we introduce Lotus, a diffusion-based visual foundation model for dense prediction. Through systematic analysis and specifically tailored diffusion formulation, Lotus finds a way to better fit the rich visual prior from pre-trained diffusion models into dense prediction. Extensive experiments demonstrate that Lotus achieves promising performance on zero-shot depth and normal estimation with minimal training data, paving the way of various practical applications.

Table 1: **Quantitative comparison on zero-shot affine-invariant depth estimation** between Lotus and SoTA methods. The upper section lists discriminative methods, the lower lists generative ones. The best and second best performances are highlighted. **Lotus-G** outperforms all others methods while **Lotus-D** is only slightly inferior to DepthAnything. $§$indicates results revised by ourselves, following Marigold (Ke et al., 2024). $\star$denotes the method relies on pre-trained Stable Diffusion.

| Method | Training Data↓ | NYUv2 (Indoor) | | | KITTI (Outdoor) | | | ETH3D (Various) | | | ScanNet (Indoor) | | | DIODE (Various) | | | Avg Rank |
|---|---|---|---|---|---|---|---|---|---|---|---|---|---|---|---|---|---|
| | | AbsRel↓ | δ1↑ | δ2↑ | AbsRel↓ | δ1↑ | δ2↑ | AbsRel↓ | δ1↑ | δ2↑ | AbsRel↓ | δ1↑ | δ2↑ | AbsRel↓ | δ1↑ | δ2↑ | |
| DiverseDepth | 320K | 11.7 | 87.5 | - | 19.0 | 70.4 | - | 22.8 | 69.4 | - | 10.9 | 88.2 | - | 37.6 | 63.1 | - | 10.6 |
| MiDaS | 2M | 11.1 | 88.5 | - | 23.6 | 63.0 | - | 18.4 | 75.2 | - | 12.1 | 84.6 | - | 33.2 | 71.5 | - | 10.2 |
| LeRes | 354K | 9.0 | 91.6 | - | 14.9 | 78.4 | - | 17.1 | 77.7 | - | 9.1 | 91.7 | - | 27.1 | 76.6 | - | 7.8 |
| Omnidata | 12.2M | 7.4 | 94.5 | - | 14.9 | 83.5 | - | 16.6 | 77.8 | - | 7.5 | 93.6 | - | 33.9 | 74.2 | - | 7.5 |
| DPT | 1.4M | 9.8 | 90.3 | - | 10.0 | 90.1 | - | 7.8 | 94.6 | - | 8.2 | 93.4 | - | 18.2 | 75.8 | - | 5.8 |
| HDN | 300K | 6.9 | 94.8 | - | 11.5 | 86.7 | - | 12.1 | 83.3 | - | 8.0 | 93.9 | - | 24.6 | 78.0 | - | 5.3 |
| GenPercept$^{\star§}$ | 74K | 5.6 | 96.0 | 99.2 | 13.0 | 84.2 | 97.2 | 7.0 | 95.6 | 98.8 | 6.2 | 96.1 | 99.1 | 35.7 | 75.6 | 86.6 | 4.9 |
| Diffusion-E2E-FT$^{\star}$ | 74K | 5.4 | 96.5 | 99.1 | 9.6 | 92.1 | 98.0 | 6.4 | 95.9 | 98.7 | 5.8 | 96.5 | 98.8 | 30.3 | 77.6 | 87.9 | 3.6 |
| DepthAnything V2 | 62.6M | 4.5 | 97.9 | 99.3 | 7.4 | 94.6 | 98.6 | 13.1 | 86.5 | 97.5 | 4.2 | 97.8 | 99.3 | 26.5 | 73.4 | 87.1 | 3.5 |
| **Lotus-D (Ours)**$^{\star}$ | 59K | 5.1 | 97.2 | 99.2 | 8.1 | 93.1 | 98.7 | 6.1 | 97.0 | 99.1 | 5.5 | 96.5 | 99.0 | 22.8 | 73.8 | 86.2 | 3.0 |
| DepthAnything | 62.6M | 4.3 | 98.1 | 99.6 | 7.6 | 94.7 | 99.2 | 12.7 | 88.2 | 98.3 | 4.3 | 98.1 | 99.6 | 26.0 | 75.9 | 87.5 | 2.4 |
| GeoWizard$^{\star§}$ | 280K | 5.6 | 96.3 | 99.1 | 14.4 | 82.0 | 96.6 | 6.6 | 95.8 | 98.4 | 6.4 | 95.0 | 98.4 | 33.5 | 72.3 | 86.5 | 3.3 |
| Marigold$_{(LCM)}$$^{\star§}$ | 74K | 6.1 | 95.8 | 99.0 | 9.8 | 91.8 | 98.7 | 6.8 | 95.6 | 99.0 | 6.9 | 94.6 | 98.6 | 30.7 | 77.5 | 89.3 | 2.9 |
| Marigold$^{\star}$ | 74K | 5.5 | 96.4 | 99.1 | 9.9 | 91.6 | 98.7 | 6.5 | 95.9 | 99.0 | 6.4 | 95.2 | 98.8 | 30.8 | 77.3 | 88.7 | 2.1 |
| **Lotus-G (Ours)**$^{\star}$ | 59K | 5.4 | 96.8 | 99.2 | 8.5 | 92.2 | 98.4 | 5.9 | 97.0 | 99.2 | 5.9 | 95.7 | 98.8 | 22.9 | 72.9 | 86.0 | 1.3 |

Table 2: **Quantitative comparison on zero-shot surface normal estimation** between Lotus and SoTA methods. Discriminative methods are shown in the upper section, generative methods in the lower. Both **Lotus-D** and **Lotus-G** outperform all other methods. $‡$refers the Marigold normal model as detailed in this link. $\star$denotes the method relies on pre-trained Stable Diffusion.

| Method | Training Data↓ | NYUv2 (Indoor) | | | ScanNet (Indoor) | | | iBims-1 (Indoor) | | | Sintel (Outdoor) | | | OASIS (Various) | | | Avg. Rank |
|---|---|---|---|---|---|---|---|---|---|---|---|---|---|---|---|---|---|
| | | m.↓ | 11.25°↑ | 30°↑ | m.↓ | 11.25°↑ | 30°↑ | m.↓ | 11.25°↑ | 30°↑ | m.↓ | 11.25°↑ | 30°↑ | m.↓ | 11.25°↑ | 30°↑ | |
| OASIS | 110K | 29.2 | 23.8 | 60.7 | 32.8 | 15.4 | 52.6 | 32.6 | 23.5 | 57.4 | 43.1 | 7.0 | 35.7 | - | - | - | 7.8 |
| Omnidata | 12.2M | 23.1 | 45.8 | 73.6 | 22.9 | 47.4 | 73.2 | 19.0 | 62.1 | 80.1 | 41.5 | 11.4 | 42.0 | 24.9 | 31.0 | 71.4 | 5.9 |
| EESNU | 2.5M | 16.2 | 58.6 | 83.5 | - | - | - | 20.0 | 58.5 | 78.2 | 42.1 | 11.5 | 41.2 | 27.7 | 24.0 | 66.6 | 5.8 |
| GenPercept$^{§\star}$ | 74K | 18.2 | 56.3 | 81.4 | 17.7 | 58.3 | 82.7 | 18.2 | 64.0 | 82.0 | 37.6 | 16.2 | 51.0 | 26.3 | 26.9 | 71.1 | 4.9 |
| Omnidata V2 | 12.2M | 17.2 | 55.5 | 83.0 | 16.2 | 60.2 | 84.7 | 18.2 | 63.9 | 81.1 | 40.5 | 14.7 | 43.5 | 24.2 | 27.7 | 74.2 | 4.4 |
| DSINE | 160K | 16.4 | 59.6 | 83.5 | 16.2 | 61.0 | 84.4 | 17.1 | 67.4 | 82.3 | 34.9 | 21.5 | 52.7 | 24.4 | 28.8 | 72.0 | 3.1 |
| Diffusion-E2E-FT$^{§\star}$ | 74K | 16.5 | 60.4 | 83.1 | 14.7 | 66.1 | 85.1 | 16.1 | 69.7 | 83.9 | 33.5 | 22.3 | 53.5 | 23.2 | 29.4 | 74.5 | 1.9 |
| **Lotus-D (Ours)**$^{\star}$ | 59K | 16.2 | 59.8 | 83.9 | 14.7 | 64.0 | 86.1 | 17.1 | 66.4 | 83.0 | 32.3 | 22.4 | 57.0 | 22.3 | 31.8 | 76.1 | 1.4 |
| Marigold$^{‡\star}$ | 74K | 20.9 | 50.5 | - | 21.3 | 45.6 | - | 18.5 | 64.7 | - | - | - | - | - | - | - | 3.6 |
| GeoWizard$^{§\star}$ | 280K | 18.9 | 50.7 | 81.5 | 17.4 | 53.8 | 83.5 | 19.3 | 63.0 | 80.3 | 40.3 | 12.3 | 43.5 | 25.2 | 23.4 | 68.1 | 3.1 |
| StableNormal$^{§\star}$ | 250K | 18.6 | 53.5 | 81.7 | 17.1 | 57.4 | 84.1 | 18.2 | 65.0 | 82.4 | 36.7 | 14.1 | 50.7 | 26.5 | 23.5 | 68.7 | 2.1 |
| **Lotus-G (Ours)**$^{\star}$ | 59K | 16.5 | 59.4 | 83.5 | 15.1 | 63.9 | 85.3 | 17.2 | 66.2 | 82.7 | 33.6 | 21.0 | 53.8 | 22.7 | 29.4 | 75.8 | 1.0 |

Table 3: **Ablation studies** on the step-by-step design of our adaptation protocol for fitting pre-trained diffusion models into dense prediction. Here we show the results in monocular depth estimation.

| Method | Training Data | NYUv2 (Indoor) | | | KITTI (Outdoor) | | | ETH3D (Various) | | | ScanNet (Indoor) | | |
|---|---|---|---|---|---|---|---|---|---|---|---|---|---|
| | | AbsRel↓ | δ1↑ | δ2↑ | AbsRel↓ | δ1↑ | δ2↑ | AbsRel↓ | δ1↑ | δ2↑ | AbsRel↓ | δ1↑ | δ2↑ |
| Direct Adaptation | 39K | 11.551 | 87.692 | 96.122 | 20.164 | 70.403 | 90.996 | 19.894 | 76.464 | 87.960 | 15.726 | 78.885 | 93.651 |
| $+ x_0$-prediction | 39K | 8.332 | 92.769 | 97.941 | 17.008 | 74.969 | 93.611 | 11.075 | 87.952 | 94.978 | 10.212 | 89.130 | 97.181 |
| + Single Time-step | 39K | 5.587 | 96.272 | 99.113 | 13.262 | 83.210 | 97.237 | 7.586 | 94.143 | 97.678 | 6.262 | 95.394 | 98.791 |
| + Detail Preserver | 39K | 5.555 | 96.303 | 99.118 | 13.170 | 83.657 | 97.454 | 7.147 | 95.000 | 98.058 | 6.201 | 95.470 | 98.814 |
| + Mixture Dataset | 59K | 5.425 | 96.597 | 99.156 | 11.324 | 87.692 | 97.780 | 6.172 | 96.077 | 98.980 | 6.024 | 96.026 | 99.730 |
| ↪ − Noise Input | 59K | 5.334 | 96.729 | 99.198 | 9.334 | 92.813 | 98.795 | 6.846 | 95.290 | 98.899 | 5.982 | 96.287 | 99.087 |
| + Disparity Space (**Lotus-G**) | 59K | 5.379 | 96.736 | 99.155 | 8.521 | 92.206 | 98.374 | 5.878 | 97.024 | 99.233 | 5.925 | 95.727 | 98.839 |
| ↪ − Noise Input (**Lotus-D**) | 59K | 5.123 | 97.182 | 99.134 | 8.117 | 93.097 | 98.654 | 6.147 | 96.964 | 99.077 | 5.494 | 96.534 | 99.039 |

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
