# Supplementary Materials of
# Lotus: Diffusion-based Visual Foundation Model for High-quality Dense Prediction

**Jing He[1]\* Haodong Li[1]\* Wei Yin[2] Yixun Liang[1] Leheng Li[1] Kaiqiang Zhou[3] Hongbo Zhang[3]
Bingbing Liu[3] Yingcong Chen[1,4]✉**
[1]HKUST(GZ) [2]University of Adelaide [3]Noah's Ark Lab [4]HKUST
{jhe812, hli736}@connect.hkust-gz.edu.cn; yingcongchen@ust.hk

## A  Experimental Settings

### A.1  Implementation Details

We implement Lotus based on Stable Diffusion V2 (Rombach et al., 2022), with text conditioning disabled. During training, we fix the time-step $t = 1000$. To optimize the model, we utilize the standard Adam optimizer with the learning rate $3 \times 10^{-5}$. All experiments are conducted on 8 NVIDIA A800 GPUs and the total batch size is 128. For our discriminative variant, we train for 4,000 steps, which takes ∼8.1 hours, while for the generative variant, we extend training to 10,000 steps, requiring ∼20.3 hours.

### A.2  Evaluation Datasets and Metrics

**Evaluation Datasets.** ① For affine-invariant depth estimation, we evaluate on 4 real-world datasets that are not seen during training: NYUv2 (Silberman et al., 2012) and ScanNet (Dai et al., 2017) all contain images of indoor scenes; KITTI (Geiger et al., 2013) contains various outdoor scenes; ETH3D (Schops et al., 2017), a high-resolution dataset, containing both indoor and outdoor scenes. ② For surface normal prediction, we employ 4 datasets for evaluation: NYUv2 (Silberman et al., 2012), ScanNet (Dai et al., 2017), and iBims-1 (Koch et al., 2018) contain real indoor scenes; Sintel (Butler et al., 2012) contains highly dynamic outdoor scenes.

**Metrics.** ① For affine-invariant depth, we follow the evaluation protocol from (Ranftl et al., 2020; Ke et al., 2024; Yang et al., 2024a;b), aligning the estimated depth predictions with available ground truths using least-squares fitting. The accuracy of the aligned predictions is assessed using the *absolute mean relative error* (AbsRel), *i.e.*, $\frac{1}{M}\sum_{i=1}^{M}|a_i - d_i|/d_i$, where $M$ is the total number of pixels, $a_i$ is the predicted depth map and $d_i$ represents the ground truth. We also report $\delta 1$ and $\delta 2$, the proportion of pixels satisfying $\text{Max}(a_i/d_i, d_i/a_i) < 1.25$ and $< 1.25^2$ respectively.

② For surface normal, following (Bae & Davison, 2024; Ye et al., 2024), we evaluate the predictions of Lotus by measuring the mean angular error for pixels with available ground truth. Additionally, we report the percentage of pixels with an angular error below $11.25°$ and $30°$.

For all tasks, we report the *Avg. Rank*, which indicates the average ranking of each method across various datasets and evaluation metrics. A lower value signifies better overall performance.

## B  Details of Direct Adaption

As illustrated in Fig. 4 of the main paper, our Direct Adaption means directly adapting the standard diffusion formulation for image generation into dense prediction task with minimal modifications. Specifically, starting with the pre-trained Stable Diffusion model, image $\mathbf{x}$ and annotation $\mathbf{y}$ are encoded using the pre-trained VAE encoder. Noise is added to the encoded annotation to obtain the noisy annotation $\mathbf{z}_t^{\mathbf{y}}$ at noise level $t \in [1, T]$. The encoded image $\mathbf{z}^{\mathbf{x}}$ is then concatenated with the noisy annotation $\mathbf{z}_t^{\mathbf{y}}$ to form the input of the denoiser U-Net model. To handle this concatenated input, the U-Net input layer is duplicated (from 4 channels to 8 channels) and its original weights are halved as initialization, which prevents activation inflation (Ke et al., 2024). Direct Adaptation is a standard multi-step formulation and optimized using the standard diffusion objective, $\epsilon$-prediction,

Table A: Experiments based on Marigold $w/$ AMRN.

| Index | Method | NYUv2 | | KITTI | |
|-------|--------|-------|-------|-------|-------|
| | | AbsRel↓ | $\delta 1\uparrow$ | AbsRel↓ | $\delta 1\uparrow$ |
| 1-1 | $\epsilon$-pred. | 6.746 | 95.021 | 11.827 | 87.065 |
| 1-2 | $\epsilon$-pred. + single step | 6.691 | 94.552 | 13.395 | 76.269 |
| 1-3 | $\epsilon$-pred. + single step + detail preserver | 6.547 | 94.772 | 12.815 | 77.829 |
| 2-1 | $v$-pred. | 6.358 | 95.188 | 10.796 | 89.726 |
| 2-2 | $v$-pred. + single step | 5.499 | 96.415 | 11.132 | 88.520 |
| 2-3 | $v$-pred. + single step + detail preserver | 5.422 | 96.517 | 10.761 | 89.826 |
| 3-1 | $x_0$-pred. | 6.262 | 95.501 | 10.769 | 89.643 |
| 3-2 | $x_0$-pred. + single step | 5.495 | 96.431 | 11.237 | 88.457 |
| **3-3** | $x_0$-**pred. + single step + detail preserver** | **5.418** | **96.542** | **10.651** | **89.887** |

Table B: Experiments based on Marigold $w/o$ AMRN.

| Index | Method | NYUv2 | | KITTI | |
|-------|--------|-------|-------|-------|-------|
| | | AbsRel↓ | $\delta 1\uparrow$ | AbsRel↓ | $\delta 1\uparrow$ |
| 1-1 | $\epsilon$-pred. | 13.110 | 85.083 | 17.655 | 75.581 |
| 1-2 | $\epsilon$-pred. + single step | 6.605 | 94.583 | 13.406 | 76.298 |
| 1-3 | $\epsilon$-pred. + single step + detail preserver | 6.582 | 94.768 | 12.823 | 77.983 |
| 2-1 | $v$-pred. | 10.634 | 89.448 | 14.328 | 84.026 |
| 2-2 | $v$-pred. + single step | 5.498 | 96.562 | 11.173 | 88.314 |
| 2-3 | $v$-pred. + single step + detail preserver | 5.459 | 96.657 | 10.814 | 89.081 |
| 3-1 | $x_0$-pred. | 8.058 | 92.834 | 12.177 | 86.301 |
| 3-2 | $x_0$-pred. + single step | 5.477 | 96.615 | 11.166 | 88.640 |
| **3-3** | $x_0$-**pred. + single step + detail preserver** | **5.396** | **96.717** | **10.575** | **89.804** |

as described in Eq. 2 of the main paper. To analyze the original diffusion formulation more effectively, we avoid specialized techniques introduced in prior methods (Ke et al., 2024; Fu et al., 2024; Xu et al., 2024; Ye et al., 2024), such as annealed multi-resolution noise (AMRN).

The AMRN strategy aims to reduce the model's variance, which has a similar effect to our design, $x_0$-pred., but through a different solution. This diminishes the impact of our method. Therefore, it is preferable to validate the effect of our designs $w/o$ AMRN. We validate this claim using the Marigold codebase, both $w/$ and $w/o$ AMRN, as shown in the Tab. A and Tab. B, respectively. In Tab. B, the performance of multi-step models follows the order: $\epsilon$-pred. $< v$-pred. $< x_0$-pred. However, in Tab. A, the differences between three parameterization types are minimal, particularly the performance of $v$-pred. and $x_0$-pred. are nearly identical. This can be attributed to the influence of AMRN, which is specifically designed for multi-step diffusion models to reduce variance and enhance performance. As a result, $x_0$-pred. shows no significant difference in reducing variance compared to the other two parameterizations. In Tab. B, when the number of time-steps is reduced to one, the performance of the model improves regardless of the parameterization type used. However, in Tab. A, the effect of single-step is unstable. This unexpected phenomenon arises from the complex, multifaceted effects of AMRN when transitioning from multi-step to single-step: ① AMRN significantly improves the multi-step model, but its effect is lost when the number of time-steps is reduced to one. ② In the single-step model, convergence is easier with limited data, leading to a slight improvement in performance. However, this also leads to catastrophic forgetting, which reduces the model's ability to handle detailed areas, especially on the KITTI dataset. In both Tab. A and Tab. B, Detail Preserver further enhances the performance of single-step model, particularly on the KITTI dataset, which contains more complex and detailed areas, such as pedestrians and fences, compared to the NYUv2 dataset. In both Tab. A and Tab. B, when using a single step ($t = T$), according to $\mathbf{v}_t = \sqrt{\bar{\alpha}_T}\epsilon - \sqrt{1 - \bar{\alpha}_T}\mathbf{z}$, since $\sqrt{\bar{\alpha}_T} \approx 0$ when $t = T$, $v$-pred. becomes equivalent to $x_0$-pred. This explains why the performances of $v$-pred. and $x_0$-pred. are nearly identical in single-step, with only minor differences. In conclusion, these experiments show that AMRN, which has a similar effect to our designs but is achieved through a different solution, diminishing the impact of our proposed designs. Therefore, it is preferable to validate the effect of our designs $w/o$ AMRN. The

experiments on Marigold $w/o$ AMRN (Tab. B) validate the effectiveness of our proposed designs, as stated in our main paper, where the best protocol is $x_0$-pred. + single step + detail preserver.

## C   ANALYSIS OF "DIRECTION($\mathbf{z}_\tau^\mathbf{y}$)" IN DDIM PROCESS (EQ. 4)

In addition to the predicted clean sample $\hat{\mathbf{z}}_\tau^\mathbf{y}$, Eq. 4 of the main paper includes another term, "direction($\mathbf{z}_\tau^\mathbf{y}$)". It is calculated according to different parameterization types:

$$\epsilon\text{-prediction: } d = w_\tau \cdot f_\theta^\epsilon$$
$$x_0\text{-prediction: } d = w_\tau \cdot \left[\frac{1}{\sqrt{1-\overline{\alpha}_\tau}}(\mathbf{z}_\tau^\mathbf{y} - \sqrt{\overline{\alpha}_\tau}f_\theta^\mathbf{z})\right] \tag{A}$$

where $d$ represents the term "direction($\mathbf{z}_\tau^\mathbf{y}$)", $w_\tau = \sqrt{1-\overline{\alpha}_{\tau-1}}$ is the weight at denoising step $\tau$. And $f_\theta^\epsilon$ and $f_\theta^\mathbf{z}$ denote the model outputs for different parameterizations. For clarity, the input of the model $f_\theta$ is omitted. As shown in Eq. A, for $x_0$-prediction, when $\tau \to 1$, *i.e.*, at the end of the denoising process, the factor $\sqrt{1-\overline{\alpha}_\tau} \to 0$, which may amplify variance from $f_\theta^\mathbf{z}$. However, its influence is limited. The reasons are as follows: ① The rate of change of $\sqrt{1-\overline{\alpha}_\tau}$ from $T$ to 1 is initially slow and then accelerates. As a result, the factor remains close to 1 for most of the denoising process, only close to 0 in the final steps. ② In $x_0$-prediction, compared to the initial denoising steps, the gap between network output $f_\theta^\mathbf{z}$ and $\mathbf{z}_\tau^\mathbf{y}$ in the final steps is much weaker and gradually approaching zero. With $\sqrt{\overline{\alpha}_\tau} \to 1$ as $\tau \to 1$, we can get $\mathbf{z}_\tau^\mathbf{y} - \sqrt{\overline{\alpha}_\tau}f_\theta^\mathbf{z} \to 0$, which may also indicate the limited influence of factor $\sqrt{1-\overline{\alpha}_\tau}$.

## D   PERFORMANCE OF $v$-PREDICTION

In sec. 4.1, we discussed two basic parameterization types: $\epsilon$-prediction and $x_0$-prediction. The latest parameterization, $v$-prediction (Salimans & Ho, 2022), combines these two basic parameterizations to avoid the invalid prediction values of $\epsilon$-prediction at some timesteps for progressive distillation. Specifically, the U-Net denoiser model $f_\theta$ learns to predict the combination of added noise $\epsilon$ and the clean sample $\mathbf{z}^\mathbf{y}$: $\mathbf{v} = \sqrt{\overline{\alpha}_\tau}\epsilon - \sqrt{1-\overline{\alpha}_\tau}\mathbf{z}^\mathbf{y}$, where $\sqrt{\overline{\alpha}_\tau}^2 + \sqrt{1-\overline{\alpha}_\tau}^2 = 1$. During inference, according to the Eq. 4 of main paper, the prediction $\hat{\mathbf{z}}_\tau^\mathbf{y} = \sqrt{\overline{\alpha}_\tau}\mathbf{z}_\tau^\mathbf{y} - \sqrt{1-\overline{\alpha}_\tau}f_\theta^\mathbf{v}$, where $f_\theta^\mathbf{v}$ represents the predicted combination, striking a balance between $\epsilon$ ($\epsilon$-prediction) and $\mathbf{z}^\mathbf{y}$ ($x_0$-prediction). As shown in Fig. A, we conduct experiments based on the settings in Fig. 5 and 6 of the main paper. The results indicate that

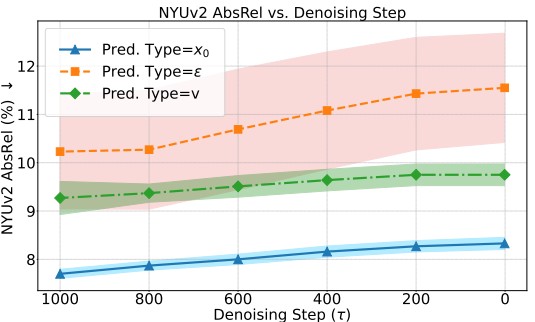

Figure A: **Quantitative evaluation of the predicted depth maps $\hat{\mathbf{z}}_\tau^\mathbf{y}$ along the denoising process.** The experimental settings are same as Fig. 5 and 6. Six steps are selected for illustration. The banded regions around each line indicate the variance, wider areas representing larger variance.

the performance of $v$-prediction falls between that of $x_0$-prediction and $\epsilon$-prediction, with moderate variance. However, for dense prediction tasks, minimizing variance is crucial to avoid unstable prediction. Therefore, $v$-prediction may not be the optimal choice. In contrast, $x_0$-prediction achieves the best performance with the lowest variance, which is why we replace the standard $\epsilon$-prediction with the more suitable $x_0$-prediction.

## E   EXPERIMENTS ON MORE DENSE PREDICTION TASKS: SEMANTIC SEGMENTATION AND DIFFUSE REFLECTANCE

To validate the generalization ability of our method on other dense prediction tasks, we further train it on semantic segmentation and diffuse reflectance prediction. Both tasks are trained using the training set of the Hypersim dataset (Roberts et al., 2021) and evaluated on their corresponding test sets. For semantic segmentation, we report the mean intersection over union (mIoU) and mean

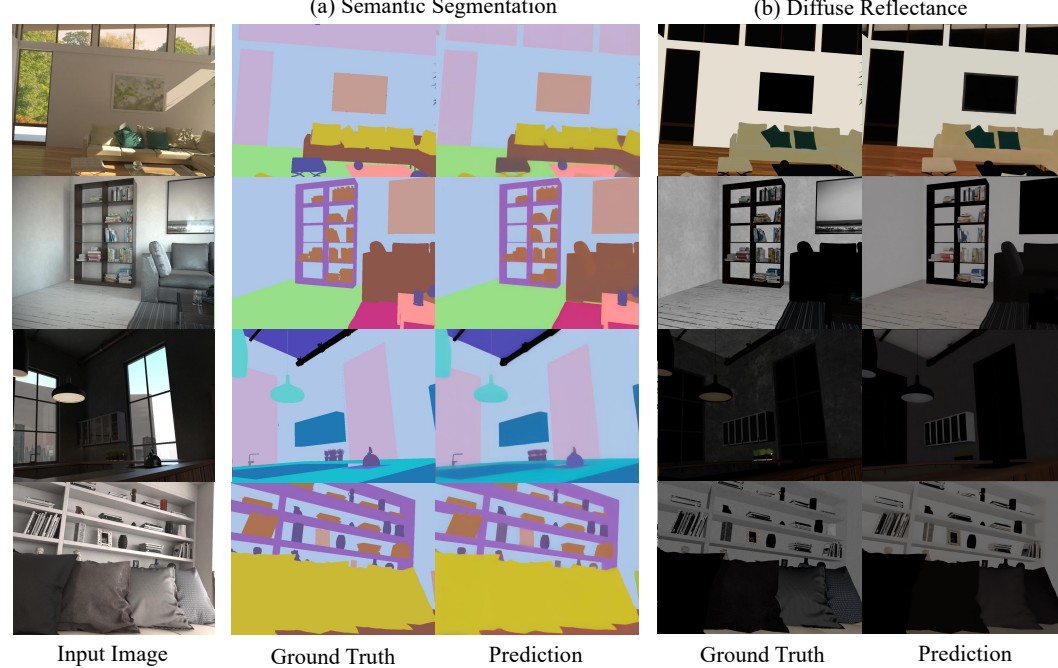

(a) Semantic Segmentation      (b) Diffuse Reflectance

Input Image    Ground Truth    Prediction    Ground Truth    Prediction

Figure B: **Experiments of Lotus on (a) semantic segmentation and (b) diffuse reflectance.** The high-quality results indicate that our method, even without task-specific designs, can be effectively applied not only to geometric dense prediction tasks, but also to semantic dense prediction tasks.

Table C: The quantitative results of semantic segmentation on Hypersim (Roberts et al., 2021) testing set. Mean values are reported from 10 independent runs.

| Method | mIoU ↑ | mAcc ↑ |
|---|---|---|
| Direct Adaption | 14.1 | 61.3 |
| **Lotus-G** | **21.2** | **65.6** |

Table D: The quantitative results of diffuse reflectance prediction on Hypersim (Roberts et al., 2021) testing set. Mean values are reported from 10 independent runs.

| Method | L1 ↓ | L2 ↓ |
|---|---|---|
| Direct Adaption | 0.198 | 0.206 |
| **Lotus-G** | **0.109** | **0.135** |

accuracy (mAcc). For diffuse reflectance prediction, we evaluate using the L1 and L2 distances to the ground truth. To enable fast evaluation, we randomly select 500 paired testing samples. In our experiments, we do not redesign any specific modules or loss functions for these tasks and maintain the original training protocol of Lotus unchanged. As shown in Tab. C and Tab. D, we compare our method with the baseline, Direct Adaption (Fig. 4 in the main paper), to assess its effectiveness. The results show that our method outperforms the baseline across all metrics. Additionally, we provide qualitative visualizations for these two tasks in Fig. B, demonstrating accurate and high-quality results. Both the quantitative and qualitative results indicate that our method, even without task-specific designs, can be effectively applied not only to geometric dense prediction tasks, as shown in the main paper, but also to semantic dense prediction tasks.

## F    FREQUENCY DOMAIN ANALYSIS OF THE DETAIL PRESERVER TAKE MONOCULAR DEPTH ESTIMATION AS AN EXAMPLE

We use fast Fourier transform (FFT) to compute the Discrete Fourier Transform (DFT) of the input images and depth map estimations with and without Detail Preserver. The entire 2D frequency domains are divided into 8 frequency groups exponentially using the base of 2, *i.e.*, the first group covers the 2D frequency map in a circle with a radius of 2, the second group covers the annular region

with radii from 2 to 4, the third group covers radii from 4 to 8, and so on. This exponential grouping allows us to analyze the frequency components across progressively larger ranges, capturing both low-frequency and high-frequency characteristics.

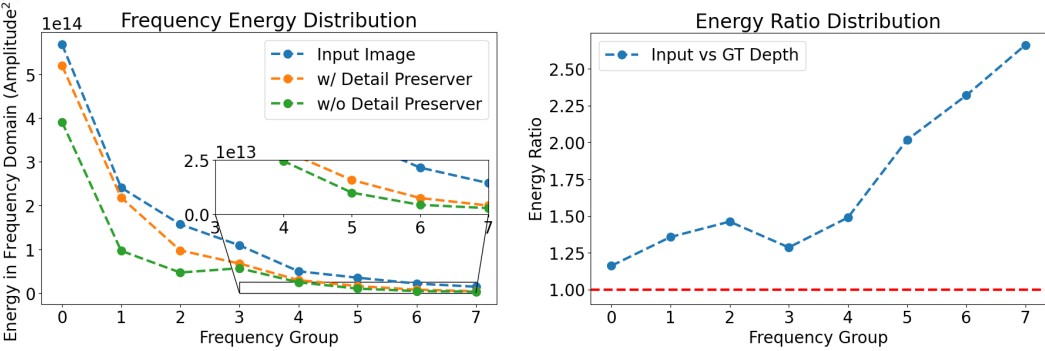

(a) Frequency domain energy distribution comparisons among input image, and depth estimations *w/* and *w/o* Detail Preserver.

(b) Frequency energy ratio **between input image and GT depth.**

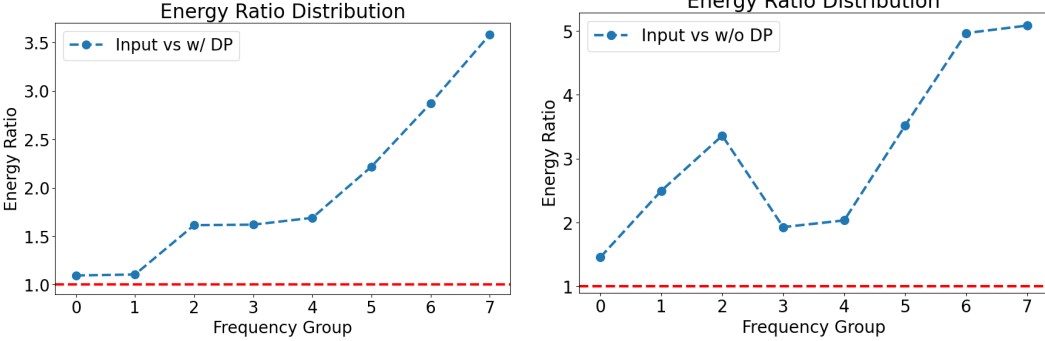

(c) Frequency energy ratio **between input image and depth estimations *w/* Detail Preserver.**

(d) Frequency energy ratio **between the input image and depth estimations *w/o* Detail Preserver.**

Figure C: **Frequency Domain Analysis of the Detail Preserver** We use Hypersim (Roberts et al., 2021) dataset to transfer the input image and depth estimation *w/* and *w/o* Detail Preserver into 2D frequency domains, using FFT. 100 pairs of {input image, depth estimation *w/* Detail Preserver, depth estimation *w/o* Detail Preserver} are randomly selected for this frequency domain analysis. Hypersim is a photorealistic synthetic dataset. Not only can Hypersim offer dense GT labels without `None` areas (which is important during FFT), its depth annotations are much fine-grained compared with real-world datasets like NYUv2 (Silberman et al., 2012) and KITTI Geiger et al. (2013).

In order to more clearly demonstrate the effect of our proposed Detail Preserver, we first analysis the experiments using Hypersim (Roberts et al., 2021) dataset to display the difference in frequency domain energy between the details from both geometry and texture (the input images); and the details from purely the geometry (the GT depth maps). As shown in Fig. Cb, the frequency domain energy between the input images and the depth annotations are plotted. Clearly we can see that the input images has much higher frequency energy in high-frequency areas, *i.e.*, group 4, 5, 6, and 7, indicating that the details in surface textures mainly contribute to high-frequency energy; while the details in geometries, which can be expressed by depth maps, are mainly concentrated into (relative) middle and low frequency areas, *i.e.*, group 0, 1, 2, and 3.

As shown in Fig. Ca , collaborating with the Detail Preserver effectively drag the frequency domain energy of depth estimation to the input image, especially on middle and low frequency domains, *i.e.*, the frequency group 0, 1, 2 and 3, highlighting the Detail Preserver's effectiveness in enhancing the geometrical details that should be reflected into depth predictions, like the fences around roads and houses (Fig. 8 of our main paper).While for high-frequency components, *i.e.*, the frequency group 4, 5, 6, and 7, which may be primarily caused by the highly detailed textures, like the signs

on the road and patterns on house surfaces, the energy in these areas between depth estimations with and without Detail Preserver is quite similar, indicating that the Detail Preserver does not copy this high-frequency and geometry-independent texture.

By comparing Fig. Cb, Cc and Cd together, we can see that Detail Preserver effectively enhances the details of geometries. This insight is evident by this phenomenon: the frequency domain energy ratio between input and depth estimation *w/* Detail Preserver, is closer to the frequency domain energy ratio between input and GT depth, compared with the frequency domain energy ratio between input and depth estimation *w/o* Detail Preserver.

## G    THE EFFECT OF DIFFERENT TIME-STEPS $t$ IN ONE-STEP DIFFUSION

In Sec. 4.2 of our main paper, we reduce the number of training time-steps of diffusion formulation to only one, and fixing the only time-step $t$ to $T$ following the diffusion formulation. In this section, we evaluate the effect of different time-steps $t$ in one-step diffusion, rather than exclusively fixing $t = T$, to validate that the rule of basic diffusion formulations should better be followed. Violating it will lead to performance degradation. As shown in Tab. E, we conduct experiments on Hypersim dataset (Roberts et al., 2021) and evaluated on NYUv2 dataset (Silberman et al., 2012), without employing the detail preserver or mixture dataset training. The results indicate that the model performs best when $t = T$ ($t = 1000$). Changing $t$ leads to a slight degradation in performance.

Table E: **The effect of different time-steps $t$ in one-step diffusion.** In this experiment, the models are trained on Hypersim dataset (Roberts et al., 2021) and evaluated on NYUv2 dataset (Silberman et al., 2012), without employing the detail preserver or mixture dataset training.

| Time-step | $t = 1000$ | $t = 750$ | $t = 500$ | $t = 250$ | $t = 1$ |
|---|---|---|---|---|---|
| AbsRel ↓ | **5.587** | 5.631 | 5.727 | 5.663 | 5.737 |
| $\delta 1$ ↑ | **96.272** | 96.165 | 96.087 | 96.141 | 96.080 |

## H    QUALITATIVE COMPARISONS

In Fig. D, we further compare the performance of our Lotus with other methods in detailed areas. The quantitative results obviously demonstrate that our method can produce much finer and more accurate depth predictions, particularly in complex regions with intricate structures, which sometimes cannot be reflected by the metrics. Also, as illustrated in Fig. E, Lotus consistently provides accurate surface normal predictions, effectively handling complex geometries and diverse environments, highlighting its robustness on fine-grained prediction.

## I    APPLICATIONS OF LOTUS

Thanks to its superiority, Lotus can seamlessly support a variety of applications. Fig. F illustrates four key applications: ① *Depth to Point Cloud.* The depth maps estimated by Lotus are projected into 3D point clouds; ② *Joint Estimation.* By incorporating a task switcher, Lotus can perform multiple tasks simultaneously, such as joint depth and normal map estimation with $100\%$ shared network parameters; ③ *Single-View Reconstruction.* Using Lotus's normal predictions, high-quality meshes can be reconstructed through through Bilateral Normal Integration (Cao et al., 2022); ④ *Multi-View Reconstruction.* Leveraging per-view depth and normal predictions from Lotus, high-quality meshes can be reconstructed with MonoSDF (Yu et al., 2022), **without RGB supervision**, showcasing Lotus's robustness and accurate spatial understanding. These applications emphasize the importance of Lotus in the field of computer vision. Its accuracy and efficiency will help in addressing increasingly complex problems.

## J    FUTURE WORK

While we have applied Lotus to two geometric dense prediction tasks, it can be seamlessly adapted to other dense prediction tasks requiring per-pixel alignment with great potential, such as panoramic

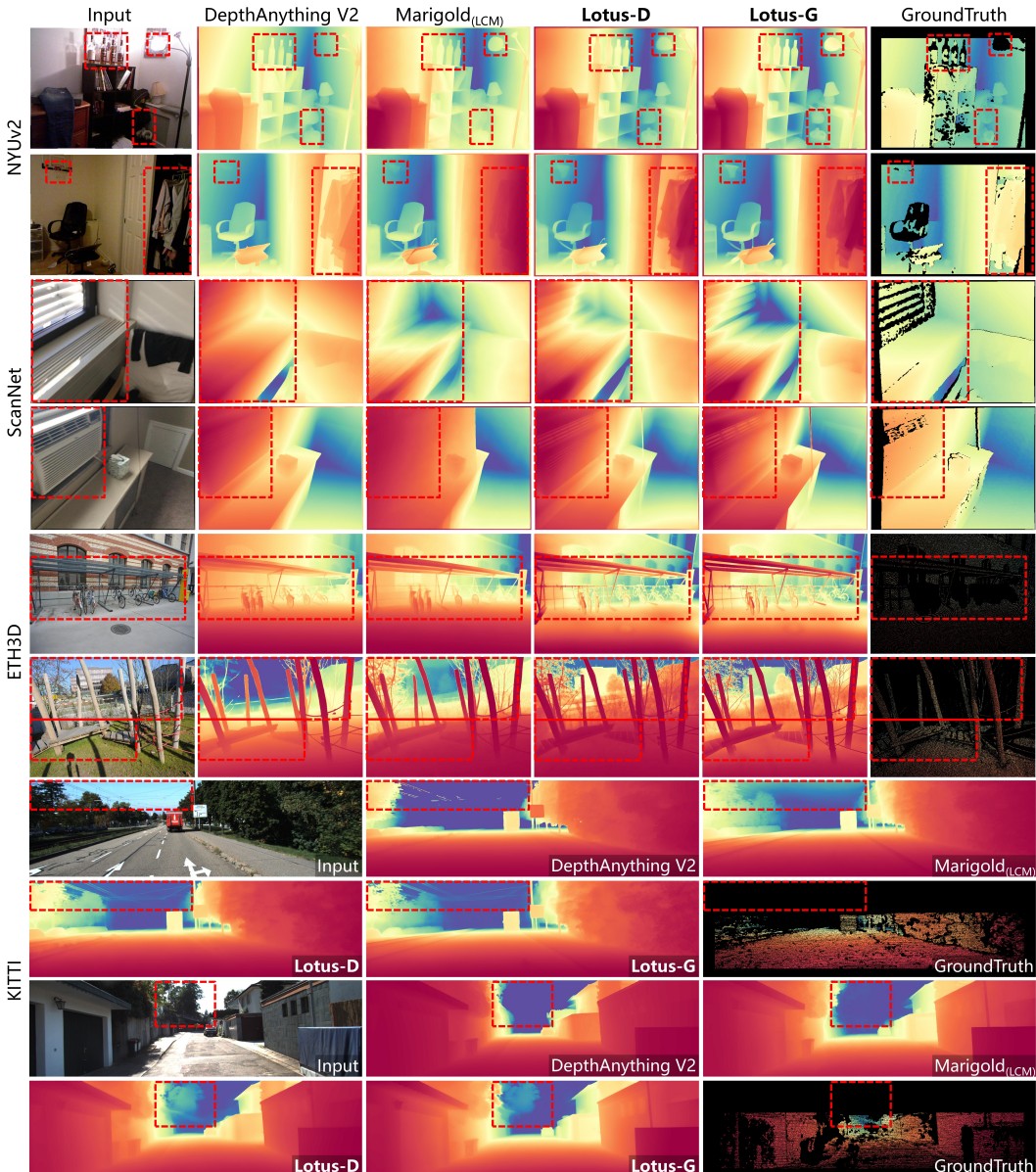

Figure D: **Qualitative comparison on zero-shot affine-invariant depth estimation.** Lotus demonstrates higher accuracy especially in detailed areas.

segmentation and image matting. Additionally, our performance is slightly behind DepthAnything (Yang et al., 2024a) which utilizes large-scale training data. In the future, scaling up the training data, as reveal in Fig. 7 and Tab. 3 ("Mixture Dataset") of the main paper, has great potential to further enhance Lotus's performance.

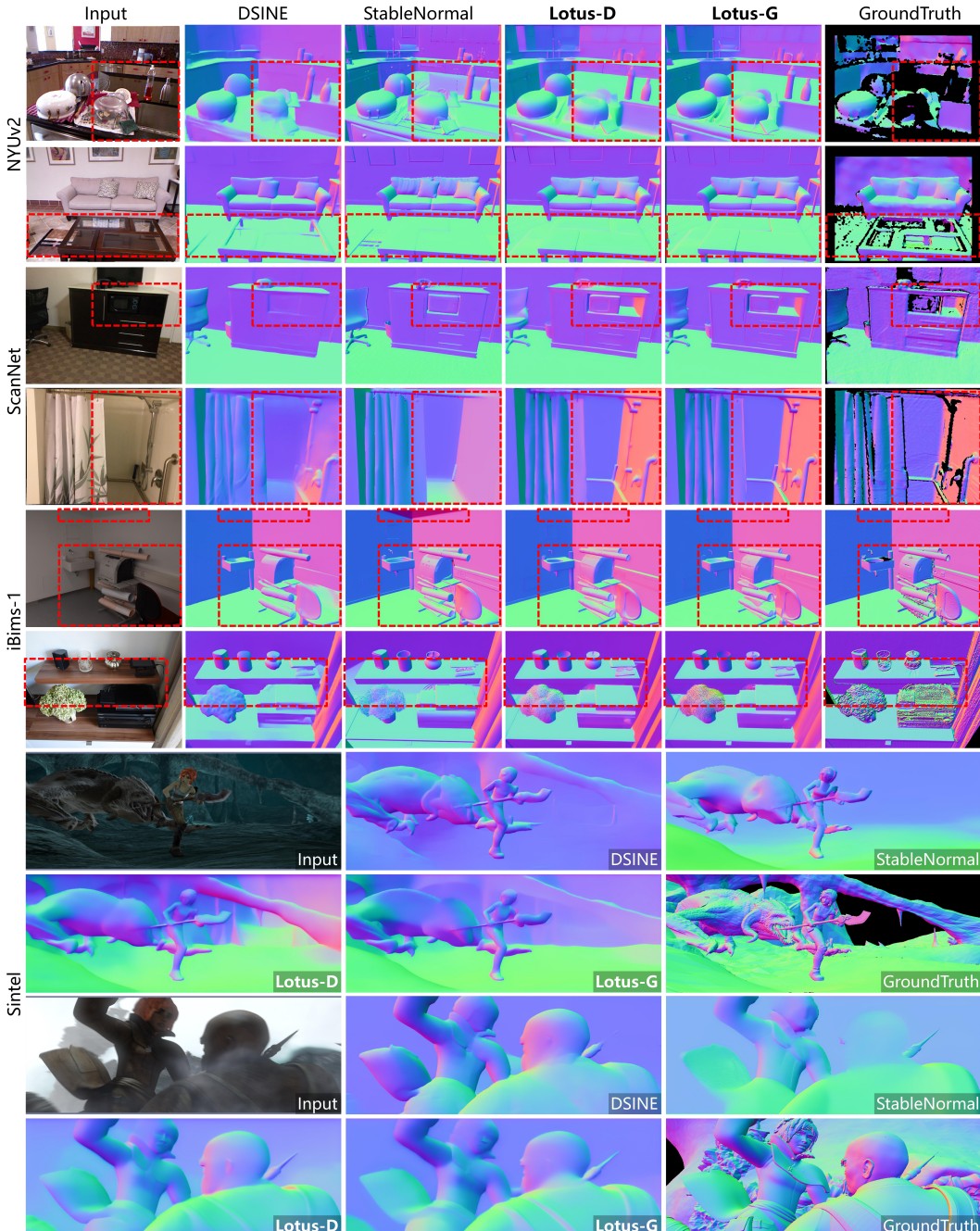

Figure E: **Qualitative comparison on zero-shot surface normal estimation.** Lotus offers improved accuracy particularly in complex regions.

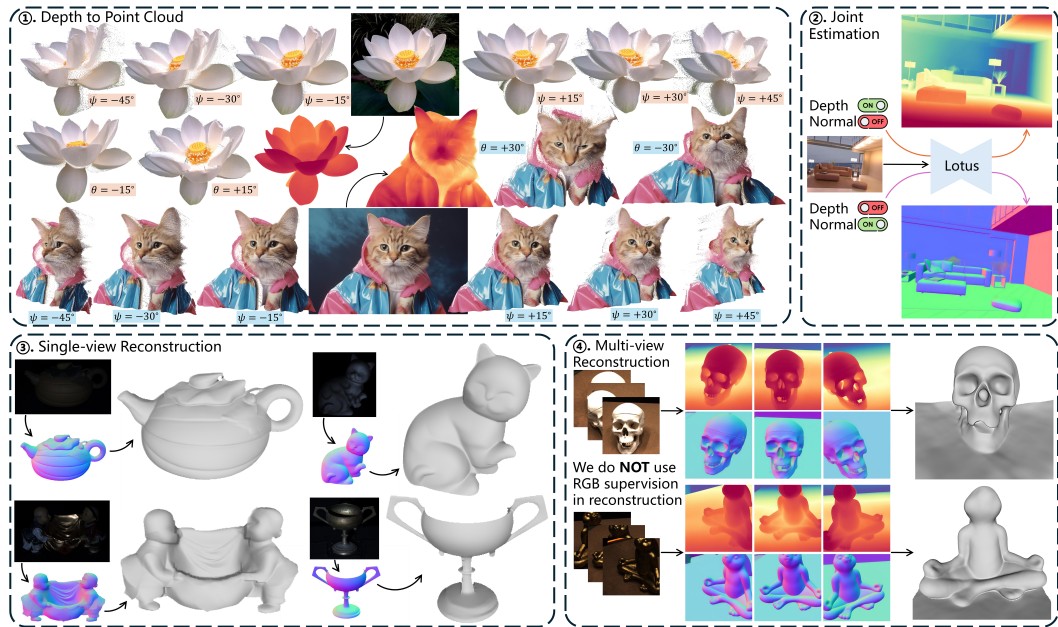

Figure F: **Applications of Lotus.** ① *Depth to 3D Point Clouds.* ② *Joint Estimation:* Simultaneous depth and normal estimation with $100\%$ shared parameters. ③ *Single-View Reconstruction:* Reconstructing 3D meshes from normal predictions. ④ *Multi-View Reconstruction:* Reconstructing high-quality meshes using depth/normal predictions **without RGB supervision**.