# OpenReview forum: "Lotus: Diffusion-based Visual Foundation Model for High-quality Dense Prediction"
_ICLR.cc/2025/Conference — ICLR 2025 Poster_

### Official Review · Reviewer_vkmx · 2024-10-29

**Soundness:** 3
**Presentation:** 3
**Contribution:** 3
**Rating:** 6
**Confidence:** 4

**Summary:**

This paper introduces a new way of using pre-trained text-to-image diffusion model for dense prediction tasks, including monocular depth estimation and surface normal estimation. Different from existing methods that use the pre-trained diffusion model without considering the dense prediction tasks, authors present several meaningful modifications. First, the parameterization type is changed in a way of predicting original image or annotation ($x_o$) instead of estimating noise. Second, the number of time-steps is reduced (even to one). Third, to enhance the capability of dealing with image details, authors utilize a task switcher that enables the proposed denoiser to generate annotation or reconstruct an input image.

**Strengths:**

1) The paper was written clearly and is easy to understand.
2) The three modifications tailored to dense prediction tasks seem to be effective.

**Weaknesses:**

1) The two tasks (depth and surface normal estimation) presented in this paper are rather insufficient. It would be better to show the possibility of using this framework in other tasks such as segmentation and detection.

2) More explanations are necessary for the detail preserver, as it is hard to understand why the image reconstruction task improves the details of estimated annotations. The image reconstruction and dense prediction tasks share the U-net denoiser, so it would be harmful to perform the two heterogeneous tasks within the single network due to some unexpected interferences. This part needs to be clarified with more experiments and detailed analysis.

3) It was shown experimentally that decreasing the number of denoising steps is more effective in the dense prediction tasks, but no theoretic analysis is given. For instance, what attributes of dense prediction tasks make it possible to reduce the number of denoising steps, unlike the image generation task? Also, does it work for other prediction tasks such as segmentation and detection?

Overall, this work introduces an interesting framework for dense prediction tasks based on pre-trained text-to-image diffusion model, demonstrating the effectiveness in the monocular depth estimation and surface normal estimation tasks. However, the performance should also be validated in different tasks (segmentation and detection), and more analysis on the detail preserver and the number of denoising steps are necessary.

**Questions:**

Refer to the comments in the weaknesses.

**Details Of Ethics Concerns:**

N.A.

---

> ### Author Response · Authors · 2024-11-22
> **Author Response to Reviewer vkmx**
>
> ## Q1: It would be better to show the possibility of using this framework in other tasks.
>
> Thanks for your suggestion. We have conducted additional experiments on semantic segmentation and diffuse reflectance prediction. For further details, please refer to Common Concern-1. Regarding object detection, it is not a dense prediction task, and therefore, we have not conducted experiments on it.
>
> ## Q2: More explanations are necessary for the detail preserver.
>
> Thanks for your valuable advice. We have conducted experiments from the view of frequency to explain the effect of the proposed detail preserver. **Please refer to Common Concern-2 and the Sec. G of the supplementary material for more details.**
>
> ## Q3: More analysis on number of time-steps.
>
> Thanks for your valuable feedback. As mentioned in Lines 337-344, unlike image generation, which is trained on billions of images, dense prediction tasks face challenges in obtaining high-quality paired training data.  This makes multi-step diffusion models for dense prediction harder to optimize. As shown in Fig. 7 of the main paper, when the amount of training data decreases, the performance of multi-step formulations degrades more rapidly than single-step ones, validating our assumption.
>
> The theoretical foundation can be found in Min-SNR[1] and ANT[2]. The multi-step diffusion training can be regarded as a multi-task learning problem, where the training of diffusion process contains T different tasks, each task represents an individual time-step.  It is obvious that optimizing a multi-task model is significantly more challenging than a single-task model. Consequently, under limited training data, single-step models perform better than multi-step ones.
>
> Additionally, 1) the diffusion priors already encode semantic information and spatial layouts, eliminating the need to relearn them; 2) dense prediction tasks are simpler than image generation  as they do not require generating new semantic informations (textures or colors). These reasons also explain why the multi-steps formulations is unnecessary in dense prediction tasks, single-step diffusion formulation can achieve even superior dense prediction performance with significantly accelerated inference speed.
>
> [1] Efficient Diffusion Training via Min-SNR Weighting Strategy
>
> [2] Addressing Negative Transfer in Diffusion Models

---

> ### Author Response · Authors · 2024-11-25
> **Authors' Kind Reminder to Reviewer vkmx**
>
> Dear Reviewer vkmx,
>
> Thank you for taking the time to review our paper and provide your valuable feedback. We have submitted detailed responses to your comments and suggestions, addressing the key points raised. **As the discussion is scheduled to end on November 26th, we would greatly appreciate it if you could review our responses and provide any further clarifications at your earliest convenience. Your feedback is extremely important to us.**
>
> Thank you again for your time and effort.
>
> Best,
>
> Authors of Submission 13

---

> ### Author Response · Authors · 2024-11-28
> **Authors' Sincere Thanks to Reviewer vkmx!**
>
> Dear Reviewer vkmx,
>
> Thanks so much for your time and effort in reviewing our submission! We deeply appearciate it.
>
> We highly value any opportunities to address any potential remaining concerns before the discussion closes, which might be helpful for improving the rating of this submission. Please do not hesitate to comment upon any further concerns. Your feedback is extremely valuable!
>
> Thanks,
>
> Authors of submission 13

---

### Official Review · Reviewer_d5L4 · 2024-10-30

**Soundness:** 2
**Presentation:** 2
**Contribution:** 3
**Rating:** 8
**Confidence:** 5

**Summary:**

The paper follows a recent trend to formulate dense per-pixel regression tasks (in particular, monocular depth and normal estimation) as conditional image generation, so as to benefit from the strong prior of (latent) denoising diffusion models. Building on Marigold [Ke et al., 2024] and its derivative GeoWizard [Fu et al., 2024], the paper proposes the following technical improvements.
1) Training the diffusion process to predict the denoised image, rather than the noise, as commonly done for image generation; Arguing that the increased variability when predicting the noise is undesirable for regression tasks.
2) Faster and less variable one-step inference, including a deterministic variant without initial random noise.
3) Additional regularisation with the image generation task to enhance the preservation of details.

**Strengths:**

The analysis of predicting the clean image vs. predicting the noise in conditional regression tasks is interesting. It is a relevant observation that, empirically, predicting the clean output directly may perform better in terms of mean prediction error. This could contribute to a better understanding of the mechanisms in diffusion models, and may potentially be interesting also for other conditional uses of those models.

The regularisation, by switching between predicting the regression target and reconstructing the input image, is plausible. While it is really a simple implementation trick with a fairly hand-wavy explanation, it might be useful for a whole range of applications that repurpose generative models for image analysis tasks.

The paper provides further evidence for the confluence of conditional generation and discriminative prediction - the evidence is thickening in the last few months that one can move to one-step prediction and even drop the random initialisation (or replace it with a constant zero-input), effectively using a pre-trained multi-step generative model as a “training scheme” or “teacher model” for a discriminative one.

**Weaknesses:**

The paper does not go very deep in its analysis of the difference between predicting the clean target or the noise. It is a plausible and important message that clean image prediction yields lower errors for certain conditional regression tasks. But the slightly higher mean error of noise prediction could, to some degree, also be an inevitable price to pay for the benefit of proper probabilistic modelling. Is the larger variability really just an algorithmic artefact, or is it perhaps a correct rendition of the underlying predictive uncertainty in the ill-posed task? The findings in the paper are an interesting starting point, but a deeper and more neutral analysis would be helpful.


The findings regarding one-step inference, as well as deterministic prediction without initial noise, are credible and important for the further development of the field. That being said, they should be discussed in the context of other recent literature, in particular [Garcia et al., 2024]. Yes, that paper only came out as a preprint 2 weeks before the ICLR deadline. I am aware that it does not qualify as "prior art", and I am not suggesting that there would be any novelty issue. But I would opine that, in this specific case, the work of Garcia et al. should be discussed and not ignored altogether. The findings - one-step training and inference, zero-noise initialisation - are very similar to one of the present paper's main messages. To maximally benefit the research community it will be important to paint a complete picture. Importantly, Garcia et al. show that these improvements are equally possible with the standard formulation (with noise prediction), after fixing a small but impactful bug in the DDIM scheduler of the most popular implementation (HuggingFace Diffusers). Again: I applaud the authors of the present paper for independently confirming that 1-step and deterministic inference are possible based on a pre-trained diffusion model. Still, since [Garcia et al.] is in some sense a bugfix and shows that these improvements apply to pretty much all of the conditional diffusion schemes of the past year if it were not for an incorrect implementation of the scheduler, I really think the authors should relate, compare and discuss that work in the final version of the paper.
- [Garcia et al., 2024] Garcia, GM, Abou Zeid, K, Schmidt, C, de Geus, D, Hermans, A, Leibe, B (2024). Fine-tuning image-conditional diffusion models is easier than you think. arXiv:2409.11355
(for the record: I am not an author of it)

My main technical criticism, where I really hope the authors could provide some insight during the discussion, comes from the ablation study in Table 3. According to that table, the “direct adaptation” of the diffusion model works a lot worse (2-3 times higher AbsRel), and both x0-prediction and single-timestep inference are needed to reach good performance. But the baselines Marigold and GeoWizard both reach comparable performance (NYU ~5.5, KITTI ~11, ETH3D ~6.5, ScanNet ~6.5) without those modifications, just doing “direct adaptation”. Why is the Lotus version of “direct adaptation” so much worse than prior art, and would the modifications have the same impact if one started from one of the competitors that already achieves low errors with direct adaptation?

I also find the "league table mentality" in the experiments (which the paper shares with many recent ones) really irritating. I would urge the authors to take a more objective, scientific stance and interpret the experimental results in a sensible and candid manner instead of desperately looking for a “win”. My interpretation of the bottom 4 lines in Table 1 is not “Lotus-G is the best”, but rather “all methods based on Stable Diffusion have comparable performance”. And it is totally ok to say that - your method is still fast, elegant and efficient in terms of training data and brings interesting insights, there is no need to over-interpret differences of mostly <0.5 percent points (mischievously, one could say that the only significant differences are on KITTI, where Lotus does not win - but then again that dataset is problematic anyway and should not be taken too serious).

The language and formulations are sometimes slightly imprecise or awkward, perhaps this can still be improved. Nothing serious that would seriously impair understanding, but still. Random examples:
- “first, their is a pair of autoencoders E(.), D(.)”. I would argue that there is only a single auto-encoder: encoder and decoder parts together form one auto-encoder, not a pair.
- “the type of parameterization is a vital configuration”. What exactly does this mean? The type of parametrisation may be important, or the configuration of the diffusion loop may be important, but something seems to be redundant in this sentence.

(there are more such small language and style issues, please check)

**Questions:**

Most important for the rebuttal: please clarify the performance gap of "direct adaptation" in Table 3 compared to other recent conditional diffusion approaches. Why do they almost match your best performance despite not using x0-prediction and single-step inference? Or, conversely, why does Lotus not perform decently without those extensions? The answer to this question is likely to significantly influence my final rating (either way).

Why did you drop DIODE from the test datasets? I am aware that it has certain issues, but since many papers (e.g., Marigold, GeoWizard, BetterDepth, etc.) show DIODE results in addition to exactly the datasets used in the paper, it would be better to also include it, for completeness. Once more, it is not a problem if, for once, you wouldn't get the bold numbers.

Similarly, why not also include DIODE and OASIS for normals, given that it does not require any training effort and makes the comparisons more complete? Also, some numbers in Table 2 differ from those I have seen floating around (although for some methods, e.g. Marigold, there do not seem to be author-approved or peer-reviewed numbers available at the time of writing). I would recommend to double-check the literature and available code bases for the rebuttal, and again for a possible final submission, to avoid swamping the literature with preliminary and potentially contradictory numbers.

---

> ### Author Response · Authors · 2024-11-22
> **Author Response to Reviewer d5L4 (Q1-3)**
>
> ## Q1: Clarify the performance gap of "direct adaptation" in Table 3 compared to other recent conditional diffusion approaches.
>
> The "direct adaption'' in our paper is directly adapted from the stable diffusion originally for image generation, i.e., specifically, it is implemented using the diffusers library (https://github.com/huggingface/diffusers/tree/main/examples/text_to_image). Unlike methods like Marigold, our direct adaption does not introduce specialized techniques for dense prediction, such as annealed **multi-resolution noise, or test-time ensembling**. These differences are the primary reasons for the performance gap.
>
> There are two main reasons why we did not base our implementation on Marigold: 1) The Marigold open-source training code was not available when we first began our experiments. 2) We start our analysis from the most straightforward way to leverage the standard diffusion formulation into dense prediction, in order to maximize the universal applicability of our conclusions. And we were also uncertain whether these specialized designs would impact our analysis. Therefore, we kept the original designs for image generation unchanged.
>
> In response, we have implemented our designs based on Marigold, as shown in the table below.  The results are consistent with our original findings (especially on NYUv2 dataset) and even achieve better performance on NYUv2. In our experiments, Marigold (v-prediction) represents its original setting, and we did not use test-time ensembling for fast evaluation.
> | Method | Training Data  | NYUv2 AbsRel⬇️ | NYUv2 delta1⬆️ | KITTI AbsRel⬇️ | KITTI delta1⬆️ |
> | --- | --- | --- | --- | --- | --- |
> | Marigold (eps-prediction) | 74K | 6.1 | 96.4 | 12.8 | 86.0 |
> | Marigold (v-prediction) | 74K | 5.63 | 96.4 | **11.8** | 87.6 |
> | Marigold (x_0-prediction) |  74K| 5.60 | 96.9 | 12.1 | 87.2 |
> | Marigold (x_0-prediction)+single step |74K  | 4.8 | 97.8 | 12.9 | 85.0 |
> | Marigold (x_0-prediction)+single step+detail preserver |74K  | **4.7** | **98.0** | **11.8** | **87.9** |
>
> ## Q2: Evaluating on more comprehensive datasets
>
> Thanks for your valuable advice. **In the supplementary material, we added the results on DIODE dataset for depth estimation in Tab. E, F (using true-depth space and disparity space, respectively) and the results on DIODE and OASIS datasets for normal estimation in Tab. H.**
>
> ## Q3: Some numbers in Table 2 differ from those I have seen floating around
>
> The results marked with * in Tab. 1 and Tab.2 were evaluated by ourselves using the tools provided by Marigold (https://github.com/prs-eth/Marigold/blob/main/eval.py)  and DSINE (https://github.com/baegwangbin/DSINE/blob/main/projects/dsine/test.py). **As the results of Marigold, DSINE, DepthAnything series, StableNormal can be reproduced**, we can ensure the correctness of the results for the other methods as well.  We will releases the code for the evaluation on other methods soon.

---

> ### Author Response · Authors · 2024-11-22
> **Author Response to Reviewer d5L4 (Q4-6)**
>
> ## Q4: Whether the larger variability in noise prediction is an algorithmic artifact or a reflection of inherent predictive uncertainty in ill-posed tasks.
>
> Thanks for your comment. The variance is indeed a reflection of inherent predictive uncertainty in ill-posed tasks, however, it should not compromise accuracy significantly. As shown in Fig. 6, the larger variance of noise prediction (illustrated by the banded regions around the line, where wider areas indicate greater variance) leads to higher errors, which is undesirable.
>
> Additionally, we evaluated the results using a testing-time ensemble strategy, similar to Marigold and GeoWizard, combining multiple inferences (10 samples in our experiments). The AbsRel scores on the NYUv2 dataset are presented in the table below. The testing-time ensemble strategy can be regarded as a Monte Carlo estimation, which approximates the true distribution through random sampling. As shown in the table below, even with the testing-time ensemble strategy, the performance of noise prediction remains inferior to that of $x_0$ prediction, indicating that the large variance in noise prediction does indeed impact prediction accuracy.
>
> | denoising time-step | t=1000 | t=800 | t=600 | t=400 | t=200 | t=1 |
> | --- | --- | --- | --- | --- | --- | --- |
> | $\epsilon$ prediction w/o ensemble | 10.23 | 10.27 | 10.69 | 11.08 | 11.43 | 11.55 |
> | $\epsilon$ prediction w/ ensemble | 9.52 | 9.55 | 9.96 | 10.4 | 10.68 | 10.83 |
> | $x_0$ prediction w/o ensemble | 7.02 | 7.87 | 8.01 |  8.16 | 8.27 | 8.33 |
> | $x_0$ prediction w/ ensemble | **6.93** | **7.81** | **7.92** | **8.09** | **8.23** | **8.29** |
>
> ## Q5:  Relate, compare and discuss the work Diffusion-E2E-FT
>
> Thanks for your valuable advice! **We have added the comparisons between our Lotus and Diffusion-E2E-FT [Garcia et al., 2024] on depth estimation in the Tab. I of supplementary material**. For normal estimation, since there are important differences in the training settings between Lotus and Diffusion-E2E-FT, we are re-training the Lotus-G model for a more fair comparison. We will add the comparison results once the training is finished.
>
> The main contribution of Diffusion-E2E-FT is an interesting phenomenon that Marigold and similar models used an inconsistent pairing of timestep and noise, leading to poor single-step predictions. Fixing this issue via setting the “timestep spacing” to "trailing" mode in the config of schedulers will avoid harmful leak of “GT” signals during inference [1], enabling more accurate single-step predictions.
>
> While the final pipeline and evaluation performance of **Lotus-D** and Diffusion-E2E-FT are quite similar, Lotus is developed from the systematic analysis of leveraging the standard stochastic diffusion formulation into dense prediction. It also incorporates innovations like the detail preserver to further improve the accuracy especially in detailed areas. Additionally Lotus (**Lotus-G**) is a stochastic model and, unlike the deterministic Diffusion-E2E-FT, it enables uncertainty predictions.
>
> [1] Common diffusion noise schedules and sample steps are flawed.
>
> ## Q6: Some writing problems
>
> Thanks for your valuable advice. We will continue to review and address similar issues to ensure the article is as professional and polished as possible before the final version.

---

> ### Author Response · Authors · 2024-11-25
> **Authors' Kind Reminder to Reviewer**
>
> Dear Reviewer d5L4,
>
> Thank you for taking the time to review our paper and provide your valuable feedback. We have submitted detailed responses to your comments and suggestions, addressing the key points raised. **As the discussion is scheduled to end on November 26th, we would greatly appreciate it if you could review our responses and provide any further clarifications at your earliest convenience. Your feedback is extremely important to us.**
>
> Thank you again for your time and effort.
>
> Best,
>
> Authors of Submission 13

---

> ### Comment · Reviewer_d5L4 · 2024-11-25
>
> Thank you for the response.
>
> Regarding Q1: I am afraid the new results do not really clarify things. If anything, they show that x0-prediction brings no real advantage over v-prediction. Single-step inference seems to indeed work better on NYUv2, but actually a bit worse on KITTI. On the latter, is exclusively the detail-preserver that gets you back to where the original baseline already was, whereas the same detail preserver does almost nothing on NYUv2. Control experiments (v-prediction + single-step, v-prediction without single-step but with detail-preserver) are missing. So no conclusive answer to my doubts whether the proposed extensions are still necessary and effective when starting from a properly tuned “direct adaptation”. This is not a killer argument against the paper, but it does mean the authors would have to (1) seriously tone down their claims and (2) add the results from the discussion to the paper, so that readers are aware of the open questions.
>
> Regarding Q2: thanks for adding these datasets. Is there a chance they can be added to Tables 1 and 2 in the main paper, and included in the calculation of the average ranks? It is fairly confusing for readers - and skews the ranking - if some of the results from the same set of experiments are in the main paper and some in the appendix.
>
> Regarding Q3: ok, fine. I was just a bit surprised to see Marigold normals, AFAIK that paper never mentions normal estimation.
>
> Overall, I slightly lower my rating. The paper is interesting enough to be published, but seems premature. It is unclear and not properly evaluated whether some of the claimed extensions (x0-prediction, detail-preserver) will stand and have any lasting impact. Single-step inference is probably here to stay, if only to be faster and for situations where a deterministic output is preferred. But I am a bit uncomfortable with publishing a paper where the majority of the contributions is not solidly backed up by experiments.

---

> ### Comment · Reviewer_d5L4 · 2024-11-25
>
> Regarding Q4, doesn’t that contradict your reply to Q1, where x0-prediction worked no better than v-prediction and only marginally better than eps-prediction?
>
> Q5: thank you. These results really belong in the main paper Table. Also, E2E-FT should be at least briefly discussed in the main paper, given that you agree its pipeline and performance are very similar to yours. Just to be sure, I repeat here that the two works were concurrent, E2E-FT does not detract from the novelty of your findings w.r.t. single-step inference.
>
> Q6: Beyond wording, please also pay attention to my comment regarding a balanced and scientific interpretation of the results. Don’t say “we are the best” when the numbers say “there isn’t much difference”. Don’t say all your extensions are important if your Marigold experiments (c.f. Q1 above ) suggest they maybe aren’t. Etc.

---

> ### Author Response · Authors · 2024-11-30
> **Author Response to Reviewer d5L4-1**
>
> Thanks for your valuable feedback! We highly value every opportunity to address your concerns. For clarity and your convenience, we divided your questions into two parts : 1) the explanation of the effect of the proposed designs, and 2) revisions to the paper, all revised parts of the main paper are highlighted in red.
>
> # Part 1: The explanation of the effect of the proposed designs.
>
> ## Q1. The performance gap of "Direct Adaptation" in Table 3 compared to other recent conditional diffusion approaches.
>
> We sincerely apologize that our response to Q1 still leaves some issues unresolved. Over the past few days, we carefully re-analyzed this question. **The main reason that our experiments in Q1 do not show significant advantages of our designs is the "annealed multi-resolution noise (AMRN)" strategy proposed in Marigold, which significantly enhances the performance of multi-step diffusion models for dense prediction.**
>
> As stated in section titled “Annealed multi-resolution noise” on Page 5 and section titled “Training noise” on Page 8 in the Marigold paper, AMRN offers two advantages:
>
> 1. Accelerating model convergence. While effective, the Marigold paper does not explain this in detail.
> 2. Improving the performance. As stated on Page 8 of the Marigold paper, "training with multi-resolution noise leads to more consistent predictions given different initial noise at inference time, and annealing further enhances this consistency." This means that **the AMRN strategy, proposed by Marigold for dense prediction, is designed to reduce the model’s variance, particularly in the multi-step formulation with v-prediction, thereby enhancing performance.**  Marigold also provides experiments in Sec. S2 to demonstrate this.
>
> **Since our original experiments for Q1 were based on the Marigold codebase, the AMRN diminishes the impact of our proposed designs.** Therefore, we argue that “Direct Adaption” would be better to directly adapt diffusion models originally for image generation, rather than incorporating designs specifically for depth prediction (or dense prediction), such as AMRN. Analyzing from the original diffusion formulation allows us to better understand how to more effectively and efficiently leverage diffusion priors for dense prediction from a more fundamental perspective.
>
> Following your suggestion, we revised our experiments by adding additional ablation experiments (Tab. A) and conducting experiments also based on Marigold codebase but w/o AMRN (Tab. B). We believe these new experiments can better validate the effect of our proposed designs.
>
> There are two tables:
>
> 1. Tab. A: The difference between this table and the one in our previous response to Q1 is that we now evaluate the models on the full NYUv2 and KITTI datasets. **We also apologize for not noticing earlier that the Marigold codebase only validates on subsets (100 images) for faster evaluation.** The validation configuration file in Marigold can be found here: https://github.com/prs-eth/Marigold/blob/main/config/dataset/dataset_val.yaml. **All the experimental settings of Lotus (main paper, suppl., the below Tab. A and B, etc.) strictly uses the complete set of all evaluation datasets. Moreover, both Tab. A and B are conducted under the same seed.**
> 2. Tab. B: The difference between Tab. A and Tab.B is that the experiments in Tab. A use AMRN while the experiments in Tab. B do not. Most of the settings in Tab. B are consistent with those in Tab. 3 of our paper, but there are some differences in the training setup: 1) All experiments in Tab. B are trained on a mixed dataset, whereas Tab. 3 uses the single hypersim dataset; 2) Tab. B uses a decaying learning rate, while Tab. 3 employs a constant learning rate.

---

> > ### Author Response · Authors · 2024-11-30
> > **Author Response to Reviewer d5L4-2**
> >
> > From the Tab. A and Tab. B, we can observe that:
> >
> > 1. In Tab. B, the performance of multi-step models follows the order: $\epsilon$-pred. < v-pred. < x0-pred. However, in Tab. A, the differences between three parameterization types are minimal, particularly the performance of v-pred. and x0-pred. are nearly identical. **This can be attributed to the influence of AMRN, which is specifically designed for multi-step diffusion models to reduce variance and enhance performance**. As a result, x0-pred. shows no significant difference in reducing variance compared to the other two parameterizations.
> > 2. In Tab. B, when the number of time-steps is reduced to one, the performance of the model improves regardless of the parameterization type used. However, in Tab. A, the effect of single-step is unstable. **This unexpected phenomenon arises from the complex, multifaceted effects of AMRN when transitioning from multi-step to single-step**: 1) AMRN significantly improves the multi-step model, but its effect is lost when the number of time-steps is reduced to one. 2) In the single-step model, convergence is easier with limited data, leading to a slight improvement in performance. However, this also leads to catastrophic forgetting, which reduces the model's ability to handle detailed areas, especially on the KITTI dataset.
> > 3. In both Tab. A and Tab.B, Detail Preserver further enhances the performance of single-step model, particularly on the KITTI dataset, which contains more complex and detailed areas, such as pedestrians and fences, compared to the NYUv2 dataset.
> > 4. In both Tab. A and Tab. B, when using a single step ($t=T$), according to $\mathbf{v}_t = \sqrt{\bar{\alpha}_T} \boldsymbol{\epsilon} - \sqrt{1 - \bar{\alpha}_T} \mathbf{z}$, since $\sqrt{\bar{\alpha}_T} \approx 0$ when $t = T$, v-pred. becomes equivalent to x0-pred. This explains why the performances of v-pred. and x0-pred. are nearly identical in single-step, with only minor differences.
> >
> > In conclusion, **these experiments show that AMRN, which seems have a similar effect to our designs but achieves this through a different solution, diminishes the impact of our proposed designs.** Therefore, it is preferable to validate the effect of our designs **w/o** AMRN when using the Marigold codebase. **The experiments on Marigold without AMRN (Tab. B) validate the effectiveness of our proposed designs, as stated in our main paper, where the best protocol is x0-pred. + single-step + detail preserver.**
> >
> > Tab. A: Experiments based on Marigold with AMRN
> >
> > | Index | Method | NYUv2 AbsRel⬇️ | NYUv2 delta1⬆️ | KITTI AbsRel⬇️ | KITTI delta1⬆️ |
> > | --- | --- | --- | --- | --- | --- |
> > | [1]-1 | Marigold (eps-prediction) | 6.746 | 95.021 | 11.827 | 87.065 |
> > | [1]-2 | Marigold (eps-prediction)+single step | 6.691 | 94.552 | 13.395 | 76.269 |
> > | [1]-3 | Marigold (eps-prediction)+single step+detail preserver | 6.547 | 94.772 | 12.815 | 77.829 |
> > | [2]-1 | Marigold (v-prediction) | 6.358 | 95.188 | 10.796 | 89.726 |
> > | [2]-2 | Marigold (v-prediction)+single step | 5.499 | 96.415 | 11.132 | 88.520 |
> > | [2]-3 | Marigold (v-prediction)+single step+detail preserver | 5.422 | 96.517 | 10.761 | 89.826 |
> > | [3]-1 | Marigold (x_0-prediction) | 6.262 | 95.501 | 10.769 | 89.643 |
> > | [3]-2 | Marigold (x_0-prediction)+single step | 5.495 | 96.431 | 11.237 | 88.457 |
> > | **[3]-3** | **Marigold (x_0-prediction)+single step+detail preserver** | **5.418** | **96.542** | **10.651** | **89.887** |
> >
> > Tab. B: Experiments based on Marigold without AMRN
> >
> > | Index | Method | NYUv2 AbsRel⬇️ | NYUv2 delta1⬆️ | KITTI AbsRel⬇️ | KITTI delta1⬆️ |
> > | --- | --- | --- | --- | --- | --- |
> > | [1]-1 | Marigold (eps-prediction) | 13.110 | 85.083 | 17.655 | 75.581 |
> > | [1]-2 | Marigold (eps-prediction)+single step | 6.605 | 94.583 | 13.406 | 76.298 |
> > | [1]-3 | Marigold (eps-prediction)+single step+detail preserver | 6.582 | 94.768 | 12.823 | 77.983 |
> > | [2]-1 | Marigold (v-prediction) | 10.634 | 89.448 | 14.328 | 84.026 |
> > | [2]-2 | Marigold (v-prediction)+single step | 5.498 | 96.562 | 11.173 | 88.314 |
> > | [2]-3 | Marigold (v-prediction)+single step+detail preserver | 5.459 | 96.657 | 10.814 | 89.081- |
> > | [3]-1 | Marigold (x_0-prediction) | 8.058 | 92.834 | 12.177 | 86.301 |
> > | [3]-2 | Marigold (x_0-prediction)+single step | 5.477 | 96.615 | 11.166 | 88.640 |
> > | **[3]-3** | **Marigold (x_0-prediction)+single step+detail preserver** | **5.396** | **96.717** | **10.575** | **89.804** |

---

> > > ### Author Response · Authors · 2024-11-30
> > > **Author Response to Reviewer d5L4-3**
> > >
> > > ## Q4. Whether the larger variability in noise prediction is an algorithmic artifact or a reflection of inherent predictive uncertainty in ill-posed tasks.
> > >
> > > > Regarding Q4, doesn’t that contradict your reply to Q1, where x0-prediction worked no better than v-prediction and only marginally better than eps-prediction?
> > > >
> > >
> > > In fact, after revising the experiments for Q1, the claim “the larger variance of noise prediction leads to higher errors” does not contradict the conclusion in Tab. B. Please disregard Tab. A, as it is less accurate due to the influence of AMRN. Additionally, this has been validated in Marigold, where they propose using AMRN to significantly reduce model variance, thereby improving performance. For further details, please refer to the section titled “Annealed Multi-Resolution Noise” on Page 5, the section “Training Noise” on Page 8 of the Marigold paper, and Sec. S2 in their supplementary material.
> > >
> > > # Part 2: Revisions to the paper.
> > >
> > > ## Q2. Update the metrics on additional datasets in the main paper.
> > >
> > > > Regarding Q2: thanks for adding these datasets. Is there a chance they can be added to Tables 1 and 2 in the main paper, and included in the calculation of the average ranks? It is fairly confusing for readers - and skews the ranking - if some of the results from the same set of experiments are in the main paper and some in the appendix
> > > >
> > >
> > > **We promise we will update them in our camera-ready version, or we will absolutely withdraw our submission.**  In addition to our method, some metrics for other methods, such GeoWizard and StableNormal on OASIS dataset, are not provided in their papers. Multi-step methods like Marigold and GeoWizard require nearly a week to evaluate on large-scale datasets like OASIS, which contains 10K test samples. Therefore, due to the time-consuming evaluation,  the complete metrics cannot be provided at this stage. **We are currently working diligently on it.**
> > >
> > > ## Q3. About Marigold normals.
> > >
> > > > Regarding Q3: ok, fine. I was just a bit surprised to see Marigold normals, AFAIK that paper never mentions normal estimation.
> > > >
> > >
> > > The Marigold team has released an official demo for normal estimation (https://huggingface.co/prs-eth/marigold-normals-lcm-v0-1, this link is already provided in the caption of main paper’s Tab. 2), and we evaluated its performance using the same normal evaluation codebase as for Lotus and other baselines like DSINE and StableNormal.
> > >
> > > ## Q5. Compare with concurrent work.
> > >
> > > > Q5: thank you. These results really belong in the main paper Table. Also, E2E-FT should be at least briefly discussed in the main paper, given that you agree its pipeline and performance are very similar to yours. Just to be sure, I repeat here that the two works were concurrent, E2E-FT does not detract from the novelty of your findings w.r.t. single-step inference.
> > > >
> > >
> > > We have updated the discussion of E2E-FT in the related work section of the main paper. We also promise that the metrics of E2E-FT will be included, along with those for Q2, in the main paper.
> > >
> > > ## Q6. About wording and more balanced and scientific interpretation of the results.
> > >
> > > > Q6: Beyond wording, please also pay attention to my comment regarding a balanced and scientific interpretation of the results. Don’t say “we are the best” when the numbers say “there isn’t much difference”. Don’t say all your extensions are important if your Marigold experiments (c.f. Q1 above ) suggest they maybe aren’t. Etc.
> > > >
> > >
> > > Thanks very much for pointing out the issue with wording and the interpretation of results. We have revised the phrasing in the latest submission, and the updated text is highlighted in red.

---

> > > > ### Author Response · Authors · 2024-12-01
> > > > **Authors' Kind Reminder to Reviewer d5L4**
> > > >
> > > > Dear Reviewer d5L4 and AC, SAC members,
> > > >
> > > > We sincerely thank you for your time and effort for reviewing our submission.
> > > >
> > > > **However, as the discussion deadline approaches in less than 3 days, we would greatly appreciate it if you could carefully review our latest responses and provide further clarifications at your earliest convenience. We highly value any opportunities to address any potential remaining issues, which we believe will be helpful for a higher rating. This will be extremely important to us!**
> > > >
> > > > Thanks again for your time and effort, we deeply appearciate it!
> > > >
> > > > Best,
> > > > Authors of Submission 13

---

> > > > > ### Comment · Reviewer_d5L4 · 2024-12-01
> > > > >
> > > > > Thank you for the active discussion and additional work!
> > > > >
> > > > > I acknowledge that the authors have gone out of their way to analyse the different approaches, and have brought up a wealth of interesting evidence. Indeed, it seems that x0-prediction and AMRN are two different strategies to control the variance, with seemingly comparable effectiveness. Also the relation to E2E-FT has been clarified, again it seems that the two independent lines of work on single-step inference reach similar results and conclusions.
> > > > >
> > > > > The analysis and results produced during the rebuttal phase bring significant added value and warrant publication. I would ask the authors to add all the tables and explanations above to the supplementary material (respectively, the arXiv appendix), such that readers have access to them without screening the present discussion. I trust the authors to comply with this request and raise my score to match the new “extended” paper.

---

> ### Author Response · Authors · 2024-12-01
> **Authors' sincere gratitude to Reviewer d5L4!**
>
> Dear Reviewer d5L4,
>
> 1. Thanks so much for your extremely thorough and constructive review of our submission and responses! The additional experiments based on your comments significantly strengthened the solidness of this submission. We deeply appreciate your valuable suggestions!
>
> 2. Furthermore, please accept our sincere gratitude for your positive feedback and the significantly improved rating! We are so excited!
>
> 3. Following your latest response, we will definitely include the additional experiments and corresponding discussions in both the
> public archive version and the camera ready version (if accepted :D) of this submission. Once again, thank you for your extremely responsible and thoughtful comments!
>
> Thanks so much and have a great day!
>
> Best,
> Authors of submission 13

---

### Official Review · Reviewer_myC3 · 2024-11-03

**Soundness:** 2
**Presentation:** 3
**Contribution:** 2
**Rating:** 6
**Confidence:** 4

**Summary:**

This paper introduces a novel approach to dense prediction tasks in computer vision. The authors analyze existing diffusion formulations and identify that traditional noise prediction methods are suboptimal for dense prediction, leading to significant prediction errors. They propose Lotus, a diffusion-based model that directly predicts annotations instead of noise, simplifying the optimization process and enhancing inference speed. The model employs a single-step diffusion process and introduces a detail preserver mechanism to improve accuracy in detail-rich areas. Experimental results demonstrate that Lotus achieves state-of-the-art performance in zero-shot depth and normal estimation tasks, significantly outperforming existing methods while requiring minimal training data.

**Strengths:**

+ This paper introduces a dense prediction method that directly predicts annotations rather than using traditional noise prediction, enhancing prediction stability and enabling one-step inference.
+ The paper provides an in-depth analysis of existing diffusion models, highlighting instability issues in dense prediction and examining the relationship between prediction variance and time steps.
+ The proposed method achieves competitive performance across multiple benchmarks, rivaling state-of-the-art approaches.

**Weaknesses:**

- In this work, the diffusion model functions more like a single-step restoration network, employing a one-step strategy for both training and inference. It may be beneficial to explore using ZtZ_tZt with different ttt values in one-step training and inference.
- For the detail-preserving component, a task switcher s is selected during each training iteration. Since predicting annotations and reconstructing images are mutually exclusive tasks, the loss function should be adjusted to reflect this "either-or" relationship.

**Questions:**

What is the relationship between the proposed network and existing conditional restoration methods? Could this design be extended to other dense prediction tasks, such as segmentation?

---

> ### Author Response · Authors · 2024-11-22
> **Author Response to Reviewer myC3**
>
> ## Q1: It may be beneficial to explore using ZtZ_tZt with different ttt values in one-step training and inference.
>
> Thanks for your comment. I am a bit unclear about what you mean by "ZtZ_tZt" and "ttt." Are you referring to $Z_t$  and $t$? In response, we trained our model using different values of $t$ (even though this deviates from the diffusion formulation) on Hypersim dataset and evaluated on NYUv2 dataset. Since our model follows the diffusion formulation, which predicts the annotation starting from noise in one step, the $Z_t$ remains Gaussian noise for different values of $t$ and is concatenated with the input image. The results are shown in the table below. As observed, the model performs best when $t=T$ (t=1000). Changing t leads to a slight degradation in performance.
>
> | time-step t | t=1000 | t=750 | t=500 | t=250 | t=1 |
> | --- | --- | --- | --- | --- | --- |
> | AbsRel ⬇️ | 5.587 | 5.631 | 5.727 | 5.663 | 5.737 |
> | delta1 ⬆️ | 96.272 | 96.165 | 96.087 | 96.141 | 96.080 |
>
> ## Q2: The loss function should be adjusted to reflect this "either-or" relationship.
>
> Thanks for your advice. In fact, our model **either** predicts annotations **or** reconstructs images by using different task switchers. As shown in Eq. 6 of the main paper, we employ different task switcher $s_x$ and $s_y$ to represent this "either-or" relationship, where $s_x$ is used for reconstruction and $s_y$ for prediction.
>
> ## Q3: What is the relationship between the proposed network and existing conditional restoration methods?
>
> To the best of our knowledge, restoration methods aim to recover high-quality or nearly realistic images from low-quality or degraded inputs. There are two main differences between Lotus and restoration models:
>
> 1.  Different model outputs. Our model’s primaty outputs are generated predicted annotations, whereas restoration models aim to generate reconstructed clear images.
> 2. Different model inputs. Our model takes clear images with noise as inputs. The noise input is introduced specifically for the diffusion process. In contrast, restoration models take degraded images as inputs.
>
> ## Q4: Could this design be extended to other dense prediction tasks, such as segmentation?
>
> Yes, we can. We have conducted additional experiments on semantic segmentation and diffuse reflectance prediction. **Please refer to Common Concern-1 and Sec. F of the supplementary materials for further details.**

---

> > ### Comment · Reviewer_myC3 · 2024-11-25
> > **Response to authors**
> >
> > Thanks for the detailed response and efforts to address the concerns. Regarding Q1, my question specifically pertains to training the diffusion model with a variable approach to *t*. For instance, instead of fixing *t*, how about training the diffusion model where *t* varies across a range (e.g., t∈[1,500]) in a adaptive manner during training? Is this approach equivalent to the "Direct Adaption" setting illustrated in Fig. 4?

---

> > > ### Author Response · Authors · 2024-11-25
> > >
> > > Thank you very much for your time and detailed feedback.
> > >
> > > “Instead of fixing *t*, how about training the diffusion model where *t* varies across a range (e.g., t∈[1,500]) in a adaptive manner during training”, are you suggesting “training **a multi-step diffusion model** using various $t$ values during different iterations,” or “training **multiple one-step diffusion models**, each with a different but fixed $t$ value”?
> > >
> > > 1. If it is the former, this approach is equivalent to Direct Adaption. In Direct Adaption, the time-step $t$ is randomly selected from the range [1, T] (i.e., [1, 1000]) in each training iteration.
> > > 2. If it is the latter, our original response to Q1 addresses your concern. In Sec. 4.2 of our main paper, we reduced the number of training time-steps of diffusion formulation to only one, and fixing the only time-step $t$ to $T$ as formulated in Eq. (5). To further investigate the setup of time-step $t$, we sampled 5 time-steps at equal intervals—$t=1,250,500,750,1000$—within the range [1,T] (i.e., [1,1000]) and trained 5 diffusion models using these different time-steps. The goal of these experiments is to evaluate the effect of different $t$ values in one-step diffusion model, rather than only fixing $t=T$ ($t=1000$). The range of time-steps  [1,T] (i.e., [1,1000]) in our experiments is the same as that used in Direct Adaption.
> > >
> > >     For your convenience, we cite our original response below and also updated these experiments in Sec. J of our supplementary material.
> > >
> > >     > We trained our model using different values of $t$ (even though this deviates from the diffusion formulation) on Hypersim dataset and evaluated on NYUv2 dataset, without employing the detail preserver or mixture dataset training. Since our model follows the diffusion formulation, which predicts the annotation starting from noise in one step, the $Z_t$ remains Gaussian noise for different values of $t$ and is concatenated with the input image. The results are shown in the table below. As observed, the model performs best when $t=T$ (t=1000). Changing t leads to a slight degradation in performance.
> > >     >
> > >
> > >     | time-step t | t=1000 | t=750 | t=500 | t=250 | t=1 |
> > >     | --- | --- | --- | --- | --- | --- |
> > >     | AbsRel ⬇️ | **5.587** | 5.631 | 5.727 | 5.663 | 5.737 |
> > >     | delta1 ⬆️ | **96.272** | 96.165 | 96.087 | 96.141 | 96.080 |
> > >
> > > If you have any additional questions or need further clarification, please do not hesitate to comment. Thank you once again for your time and feedback. We look forward to hearing from you!

---

> > > > ### Comment · Reviewer_myC3 · 2024-11-28
> > > > **Response to authors**
> > > >
> > > > Thanks for authors' detailed response on my comments. Most of my concerns have been addressed by the response. As a result, I will maintain my score.

---

> ### Author Response · Authors · 2024-11-25
> **Authors' Kind Reminder to Reviewer myC3**
>
> Dear Reviewer myC3,
>
> Thank you for taking the time to review our paper and provide your valuable feedback. We have submitted detailed responses to your comments and suggestions, addressing the key points raised. **As the discussion is scheduled to end on November 26th, we would greatly appreciate it if you could review our responses and provide any further clarifications at your earliest convenience. Your feedback is extremely important to us.**
>
> Thank you again for your time and effort.
>
> Best,
>
> Authors of Submission 13

---

> ### Author Response · Authors · 2024-11-28
> **Authors' Sincere Thanks to Reviewer myC3!**
>
> Dear Reviewer myC3,
>
> Thanks so much for your prompt and kind response! We are so excited to see that!
>
> We highly value any opportunities to address any potential remaining concerns before the discussion closes, which might be helpful for improving the rating of this submission. Please do not hesitate to comment upon any further concerns. Your feedback is extremely valuable!
>
> We deeply appearciate your time and effort in reviewing our submission and writing responses.
>
> Thanks,
>
> Authors of submission 13

---

### Official Review · Reviewer_forA · 2024-11-04

**Soundness:** 2
**Presentation:** 4
**Contribution:** 2
**Rating:** 6
**Confidence:** 5

**Summary:**

This paper introduces a novel method for dense prediction tasks by fine-tuning a pre-trained diffusion model. To achieve stable and consistent predictions, the method shifts the model’s prediction from epsilon to x0 (clean image) prediction, significantly reducing the variance in outputs. This transition allows for accurate single-step inference, which simplifies the process while maintaining high-quality results.

To avoid generating excessively smoothed outputs—a common issue in dense prediction with diffusion models—the approach incorporates input frame reconstruction into the training process. This addition serves to preserve finer details and structure in the output predictions, balancing accuracy with realistic visual representation.

The proposed method is evaluated in a challenging zero-shot scenario, where its performance is benchmarked against both discriminative and generative approaches. Despite being fine-tuned on limited data, the model demonstrates competitive, promising performance, highlighting its robustness and adaptability for dense prediction tasks.

**Strengths:**

The paper presents its motivation clearly, leading readers through a coherent explanation of the research objectives and the rationale behind the proposed method. Supported by carefully crafted figures, the narrative flows in a way that helps readers follow the progression of ideas and understand the purpose of each methodological contribution. Each figure complements the text by visually illustrating the transition from epsilon prediction to x0 prediction, the integration of input frame reconstruction, and the steps involved in single-step inference. This combination of clear motivation and visual aids effectively demonstrates how the approach stabilizes predictions and reduces variance, helping readers grasp the value and impact of the proposed method for dense prediction tasks, even in zero-shot settings with limited data.

**Weaknesses:**

The proposed method, while effective in its application, appears relatively simplistic and may lack substantial innovation. Both the use of  \mathbf{x}_0  prediction and few-step inference have been previously explored in the diffusion model literature, which limits the originality of these aspects. The  \mathbf{x}_0  prediction has been extensively discussed in seminal works such as “Denoising Diffusion Probabilistic Models” and Stable Diffusion. Similarly, few-step inference has been considered in studies like “UFOGen: You Forward Once Large Scale Text-to-Image Generation via Diffusion GANs” and “One-step Diffusion with Distribution Matching Distillation.” Consequently, these elements alone may not constitute a significant contribution to the field.

Furthermore, the paper’s evaluation and analysis are narrowly focused on the zero-shot scenario, which, while valuable, provides a limited perspective on the method’s potential. To fully illustrate the strengths of this approach for generative dense prediction, a broader evaluation across diverse scenarios beyond zero-shot would enhance the analysis. Expanding the experiments to include intra-dataset inference, or testing in varying contexts, such as semantic segmentation or optical flow, could highlight the model’s generalization abilities and its versatility in handling different levels of supervision.

I also acknowledge the authors’ claims regarding the model’s performance in minimal-data contexts. Exploring the scalability of the model could offer significant insights, especially given the inherent scalability of diffusion-based methods. Demonstrating the model’s performance with larger, more complex training datasets, similar to “DepthAnything,” would facilitate a deeper understanding of its robustness and computational efficiency. This could be achieved by training the model with datasets of varying sizes and evaluating the performance gains of such a diffusion-based approach. An expanded evaluation would provide a more balanced perspective, showcasing the model’s adaptability beyond minimal-data contexts and emphasizing its potential applicability in real-world, data-rich environments.

**Questions:**

I’m curious about the mention of “removing noise” on line 430. In the figure, the noise appears to be concatenated with the input frame—how exactly is this term removed? What value, if any, is set in its place? Does this modification affect any underlying assumptions of diffusion models?

---

> ### Author Response · Authors · 2024-11-22
> **Author Response to Reviewer to forA**
>
> ## Q1: About "removing noise'' of Lotus on line 430
>
> The ``removing noise'' version of Lotus, Lotus-D, is a new model that directly inputs the 4-channel image latent into the UNet model without concatenating it with noise. The purpose of Lotus-D is just to align with the deterministic nature of traditional models and the deterministic nature of benchmarks. Unlike Lotus-G, Lotus-D does not strictly adhere to the diffusion formulation. However, its value lies in the powerful diffusion priors. Only by leveraging the diffusion priors without the full diffusion process, it is still able to achieve superior performance.
>
> ## Q2: The proposed method appears relatively simplistic and may lack substantial innovation
>
> Thanks for your comment, however, we respectfully disagree with it. We would like to emphasis that our contribution is the proposed **fine-tuning protocol for dense prediction tasks,** which not only achieves new SoTA performance but is also significantly faster than most existing diffusion-based methods. Our contribution is encouraged by reviewer myc3, d5L4, and vkmx, as our strengths. The related works mentioned in your comment are specifically for a boarder realm of  image generation, instead of image dense prediction.
>
> Additionally, as we stated in our paper, ``the standard diffusion formulation originally designed for image generation is unsuitable for dense prediction tasks''. Experimentally, we directly adapted the diffusion formulation for image generation to dense prediction task and observed poor performance (see Lines 263–289).  Therefore, it is necessary to investigate a more tailed formulation for dense prediction that can more effectively leverage diffusion priors, which is what our paper achieves.
>
> ## Q3: Expanding the experiments to include intra-dataset inference or testing in varying contexts.
>
> Thanks for your valuable suggestion. We have conduct additional experiments in two other contexts: semantic segmentation and diffuse reflectance prediction. Both tasks were evaluated in intra-dataset (in-domain) settings, and Lotus significantly outperforms the baseline in these scenarios. **Please refer to Common Concern-1 and the Sec. F of supplementary material for further details.**
>
> However for depth and normal estimation, we evaluate Lotus on zero-shot benchmarks because 1) zero-shot evaluation is more challenging and more close to the practical unknown scenarios during application, and 2) nearly all recent method, no matter in depth and normal estimation, report their results in zero-shot benchmarks.
>
> ## Q4: Training the model with datasets of varying sizes, and scaling up the training dataset.
> | # Number of training data | 5K | 10K | 19K | 39K |
> | --- | --- | --- | --- | --- |
> | AbsRel (⬇️) | 5.97 | 5.82 | 5.78 | 5.72 |
>
> 1. Please see the Fig. 7 of the main paper, we have conducted experiments using training datasets of varying sizes, with the corresponding values presented in the table below. The performance of our single-step formulation improves as the training data scales up. This experiment demonstrate that our model has the potential to benefit further from increased data scaling, with similar or better data quality.
> 2. However,  we are currently unable to scale up our training dataset to the same extent as DepthAnything, as our hardware is not capable in training large models with 60 million samples.

---

> > ### Comment · Reviewer_forA · 2024-11-25
> >
> > I appreciate the authors’ response to my questions. However, I am not fully convinced by the explanation regarding the simplicity and lack of innovation. Prediction and few-step inference have been explored previously in the context of generative models. In this work, although the authors claim the method is tailored for image dense prediction, it still follows the formulation of diffusion models. As a result, the application of these techniques appears straightforward and does not appear to offer significant novelty or contribute notably to advancing the field.
> >
> > I acknowledge that the validation of the epsilon-prediction formulation, which shows poor performance, is intuitive. Since dense prediction is a task that tends to be more deterministic, switching to x0 prediction makes the task easier for the model to learn, and the model likely does not require much diversity (as generation) in the output. However, I still feel that this contribution is more engineering-oriented, and I do not see a substantial non-trivial contribution from it.

---

> > > ### Author Response · Authors · 2024-11-30
> > > **Author Response to Reviewer to forA**
> > >
> > > We appreciate the reviewer’s feedback, but we believe there is a misunderstanding regarding the novelty and contribution of our work.
> > >
> > > ## **1. New insights and a deeper understanding are equally important.**
> > >
> > > We would like to clarify that contributions to a field do not solely lie in proposing new model structures or paradigms. New insights and a deeper understanding when a model or a paradigm is adapted to new task (i.e., from generation to perception) are equally important and can lead to meaningful advancements.
> > >
> > > ## **2. There is no clear guidance on what exactly should we do for dense prediction in an effective and efficient manner.**
> > >
> > > You mentioned that “prediction types and few-step inference have been explored in the filed of generative models”. However, **when applying diffusion models to dense prediction tasks, what exactly should we do?** Should we rely on $\epsilon$-prediction, as in existing diffusion models, or should we use $x_0$-prediction? **No clear guidance has been provided**. Similarly, in terms of efficiency, is the multi-step diffusion formulation still necessary and is it possible to simply retain the original model and apply it in a single step, rather than proposing a new model and using distillation techniques as is done in image generation? We need to analyze these key factors for dense prediction in order to enhance the model’s performance.
> > >
> > > ## **3. Our contributions**
> > >
> > > In this regard, we believe our work makes solid contributions to the field of dense prediction using diffusion priors. **Rather than simply reusing existing designs, our approach offers new insights that move beyond blindly following the diffusion model for image generation**. As a result, we propose a more tailored and efficient protocol for dense prediction.
> > >
> > > Below, we highlight the key contributions of our work:
> > >
> > > 1. Parameterization types (or “Prediction types” as mentioned above):
> > > We analyze two key parameterization types, $\epsilon$-prediction and $x_0$-prediction, both **theoretically and experimentally**, to demonstrate that $\epsilon$-prediction introduces larger variance than $x_0$-prediction. Although the large variance enhances diversity for image generation, it lead to unstable predictions in dense prediction tasks, potentially resulting in significant errors. **In contrast, existing methods, like Marigold and GeoWizard, uncritically follow the principle of diffusion models in image generation. They require additional techniques, such as multi-resolution noise and testing-time ensembling, to mitigate variance and improve accuracy.**
> > > 2. Single-step diffusion formulation
> > > We propose fine-tuning the pre-trained diffusion model with fewer training time-steps. These time-steps are obtained by evenly sampling the original set of time-steps at intervals, as shown in Eq. 5.  Through extensive experiments with different numbers of time-steps and data scales, we find that the **multi-step diffusion formulation is unnecessary for dense prediction**. Although multi-step formulation may achieve similar performance with single-step given infinite training data, as illustrated in Fig 7 of the main paper, it is extremely challenging. **In contrast, existing methods like Marigold and GeoWizard still rely on multi-step formulations, which are time-consuming.**
> > > 3. Detail Preserver
> > > To improve prediction accuracy in detailed areas, **we introduce a novel regularization strategy** called Detail Preserver. This strategy prevents the pre-trained model from losing its ability to handle fine details due to catastrophic forgetting. **We also analyze its effectiveness from a frequency perspective in Sec. G of the supplementary material and report the ablation studies in Tab. 3 of the main paper.**
> > >
> > > Overall, our method provides **new insights** into leveraging diffusion priors for dense prediction, **both theoretically and experimentally**, and introduces a simple yet effective fine-tuning protocol. We believe our work provides significant contributions to the field of dense prediction using diffusion priors.

---

> > > > ### Author Response · Authors · 2024-12-01
> > > > **Authors' Kind Reminder to Reviewer forA**
> > > >
> > > > Dear Reviewer forA and AC, SAC members,
> > > >
> > > > We sincerely thank you for your time and effort for reviewing our submission.
> > > >
> > > > **However, as the discussion deadline approaches in less than 3 days, we would greatly appreciate it if you could carefully review our latest responses and provide further clarifications at your earliest convenience. We highly value any opportunities to address any potential remaining issues, which we believe will be helpful for a higher rating. This will be extremely important to us!**
> > > >
> > > > Thanks again for your time and effort, we deeply appearciate it!
> > > >
> > > > Best,
> > > > Authors of Submission 13

---

> ### Author Response · Authors · 2024-11-25
> **Authors' Kind Reminder to Reviewer to forA**
>
> Dear Reviewer forA,
>
> Thank you for taking the time to review our paper and provide your valuable feedback. We have submitted detailed responses to your comments and suggestions, addressing the key points raised. **As the discussion is scheduled to end on November 26th, we would greatly appreciate it if you could review our responses and provide any further clarifications at your earliest convenience. Your feedback is extremely important to us.**
>
> Thank you again for your time and effort.
>
> Best,
>
> Authors of Submission 13

---

> ### Author Response · Authors · 2024-11-25
> **Authors' Second Kind Reminder to Reviewer forA**
>
> Dear Reviewer forA,
>
> I hope this message finds you well.
>
> Thank you for your time and effort for reviewing our submission. **As the discussion deadline (November 26th) approaches,** **we would greatly appreciate it if you could review our responses and provide any further clarifications at your earliest convenience.** Your feedback is extremely valuable!
>
> Thanks again for your time and effort!
>
> Best,
>
> Authors of Submission 13

---

> ### Author Response · Authors · 2024-12-01
> **Authors' Second Kind Reminder to Reviewer forA (Only 1 Day Left for Reviewers to Post a Message)**
>
> Dear Reviewer forA,
>
> We sincerely thank you for your time and effort for reviewing our submission, and writing your responses.
>
> **However, as the discussion deadline approaches in less than 2 days (only 1 day left for reviewers to post a message), we would greatly appreciate it if you could carefully review our latest responses and provide further clarifications at your earliest convenience. We highly value any opportunities to address any potential remaining issues, which we believe will be helpful for a higher rating. Your timely feedback is extremely important to us!**
>
> Once again, thanks so much for your time and effort, we deeply appearciate it!
>
> Best,
>
> Authors of Submission 13

---

### Official Review · Reviewer_kYir · 2024-11-04

**Soundness:** 2
**Presentation:** 2
**Contribution:** 2
**Rating:** 6
**Confidence:** 4

**Summary:**

The authors simply told us diffusion-based method is not suitable, not necessary, and even harm dense prediction quality. They form a story that they reduce the diffusion step to 1, while adding a regularization of detail preserver, and finetune from a large-scale UNet pretrained on T2I task, to achieve good dense prediction performance.

**Strengths:**

- The authors presented an interesting story of removing 'diffusion' from dense prediction to make things less complicated.
- If the proposed method and claims are proved to be effective, it shows using a diffusion-based dense prediction method is overkill compared to the efficiency of a 1-step direct prediction. It can be a nice finding.

**Weaknesses:**

- What is the fundamental difference between the proposed method with a model trained with UNet using latent-space? If there is no diffusion process and the model is doing x0 prediction, why it is still called diffusion method, especially when the authors proposed a discriminator version, does it just fall back to a simple UNet-based method?
- If the authors claim the powerful prediction quality can come from the strong prior of the pretrained model, but the currently proposed detail preserver, is just learning the reconstruction, and can be lazily learned via the skip connection within the UNet structure, it does not seem to preserve the original generator capability. The reviewer believe after longer training or large batch size training, the performance will degrade more. It is interesting that the detail preserver can enhance the detail by a large margin, given the reconstruction can be easily learned by the residual connection in each layer.
- If the authors claim it can be served as a visual foundation model, it needs to show more capability of generalization or potentials of other tasks, either through finetuning or adaptation.
- The qualitative results seem still not strong enough compared with other baselines, and are very close to Marigold.

**Questions:**

- Do we need to finetune the VAE or decoder for dense representation given the original VAE can be only trained on image data?
- How did the author evaluate the generative methods given the randomness?
- Does the conclusions from this paper also apply to other simpler vision tasks? How do the author feel about segmentation task? Also whether diffusion is just not suitable for any tasks requiring pixel-level preservation?

Overall I feel the authors proposed a good observation for dense prediction, but a couple of statements are over-claimed, like foundation model, 1-step 'diffusion' is enough, or detail preserver is necessary.

---

> ### Author Response · Authors · 2024-11-22
> **Author Response to Reviewer kYir (Q1-3)**
>
> ## Q1: The relationship of our proposed method between diffusion and UNet-based model.
>
> ### Q1-1: Why the proposed method is called ``diffusion model''?
>
> Lotus is called "diffusion-based'' method in our title. More precisely,  the reasons are: 1) Lotus, especially the Lotus-G, strictly follows the principle of diffusion formulation at time-step $T$ to generate annotation from noise. It contains single-step noising and denoising process. 2) Lotus, either Lotus-G or Lotus-D, relies on the diffusion priors (which is the valuable and powerful pre-trained network parameters) requiring adapting and fine-tuning under the basic diffusion pipeline. Therefore, the claim of "diffusion-based'' model is reasonable.
>
> ### Q1-2: What is the difference between the proposed method with UNet-based model?
>
> Firstly, diffusion-based model and UNet-based model are interconnected, as UNet serves as the denoiser in diffusion models.
>
> Secondly, Lotus (Lotus-G) still follows the principle of diffusion denoising formulation, the UNet denoises the concatenated latent feature of Gaussian noise and the input image, and producing the latent feature of the dense prediction. Thus, it is not merely a simple UNet-based model.
>
> Lastly, and importantly, we emphasize that our main contribution lies in exploring the most suitable diffusion formulation to effectively leveraging the powerful diffusion priors for dense prediction tasks, rather than focusing on model structure.  Through systematic analysis and extensive experiments,  we proposed a novel adaptation protocol that significantly outperforms previous diffusion-based methods in both efficiency and accuracy.
>
> ### Q1-3: Dose the discriminative version just fall back to a simple UNet-based method?
>
> Although our Lotus-D does not strictly follow the diffusion formulation, as the noise input is removed, to meet the discriminative requirements of perceptions tasks.
>
> Its value lies not in its model structure, but in demonstrating that even without the noise input in diffusion formulation, SoTA performance can still be achieved with the support of pre-trained diffusion priors. This result provides an intriguing insight for bridging the fields of generation and perception.
>
> ## Q2: The effect of Detail Preserver
>
> Thanks for your comment. In our paper, the “original detailed generation ability” does not mean the ability to generate new textures or colors like that in image generation task. It is also not desired to “imagine” new details, which may lead to deviation from input image. Instead, it means to preserve those details that should be clearly expressed in dense predictions, *e.g.*, the geometrical details for depth/normal prediction.
>
> The original diffusion model excels at generating detailed images but may lose its ability to generate intricate details with dense annotation tasks due to catastrophic forgetting. To address this, we propose a regularization strategy called *Detail Preserver*. This approach uses a switcher: when set to $s_y$, the model focuses on predicting annotation; when set to $s_x$. it reconstructs the input image to retain its detail generation ability. This dual focus improves dense predictions in intricate regions and enhances evaluation performance.
>
> **Please refer to Common Concern-2 and Sec. G of the supplementary materials for more details.**
>
> ## Q3: If the authors claim it can be served as a visual foundation model, it needs to show more capability of generalization or potentials of other tasks.
>
> Thanks for your suggestion.
>
> We have trained Lotus on two additional, representative dense prediction tasks: semantic segmentation and diffuse reflection prediction, to further support our claim. **Please refer to Common Concern-1 and Sec. F of the supplementary materials for detailed information.**

---

> ### Author Response · Authors · 2024-11-22
> **Author Response to Reviewer kYir (Q4-5)**
>
> ## Q4: Do we need to fine-tune the VAE or decoder for dense representation?
>
> We do not fine-tune the pre-trained VAE and only adapt the annotations into 3-channel input. Depth maps are repeated to 3 channels, surface normal maps (which originally are 3-channels) remain unchanged, and segmentation maps are converted to color maps.  The generalization capability of VAE on dense annotations is also validated in previous works, e.g., Marigold[1] and GeoWizard[2].
>
> [1] Repurposing diffusion-based image generators for monocular depth estimation.
>
> [2] Geowizard: Unleashing the diffusion priors for 3d geometry estimation from a
> single image.
>
> ## Q5: How did the author evaluate the generative methods given the randomness?
>
> All evaluations are performed with the same seed across Lotus and all baselines. To ensure a fairer comparison, we also report the mean and variance values of the metrics for our Lotus  over 10 independent runs, as shown in the tables below. These results show that the variance of our method is small, only approximately e-3~e-5, and the mean values across 10 runs are  consistent with those reported int the Tab. 1 and Tab. 2 of the main paper, highlighting the robustness of our method.
>
> **We have updated these experiments in Sec. H of our supplementary material for your kind reference.**
>
> The Lotus-G’s results in depth estimation, in true-depth space.
> | Dataset | AbsRel ⬇️| RMSE⬇️ | delta1⬆️ | delta2⬆️ | delta3⬆️ |
> | --- | --- | --- | --- | --- | --- |
> | diode | 0.3311 (±1.4285e-04) | 3.8836 (±1.3504e-03) | 0.7360 (±4.4283e-04) | 0.8764 (±5.4613e-04) | 0.9304 (±1.4050e-04) |
> | eth3d | 0.0617 (±1.9343e-04) | 0.5806 (±9.8765e-04) | 0.9605 (±4.3380e-04) | 0.9897 (±8.3054e-05) | 0.9957 (±7.8772e-05) |
> | kitti | 0.1134 (±5.8194e-05) | 3.5379 (±8.9554e-04) | 0.8771 (±1.3912e-04) | 0.9776 (±6.6427e-05) | 0.9930 (±4.0593e-05) |
> | nyuv2 | 0.0542 (±1.0072e-04) | 0.2220 (±1.5572e-04) | 0.9661 (±2.5504e-04) | 0.9915 (±7.9410e-05) | 0.9978 (±2.9075e-05) |
> | scannet | 0.0603 (±1.7740e-04) | 0.1597 (±3.5362e-04) | 0.9590 (±1.4926e-04) | 0.9893 (±2.3924e-04) | 0.9972 (±1.0019e-04) |
>
> The Lotus-G’s results in normal estimation.
> | Dataset | AbsRel⬇️ | RMSE⬇️ | delta1⬆️ | delta2 ⬆️| delta3⬆️ |
> | --- | --- | --- | --- | --- | --- |
> | ibims | 17.4988 (±2.7417e-02) | 30.7278 (±2.5334e-02) | 65.9255 (±1.2659e-01) | 79.0048 (±5.5158e-02) | 82.6473 (±4.3533e-02) |
> | nyuv2 | 16.9408 (±6.8807e-03) | 26.5460 (±1.0066e-02) | 59.1155 (±1.7356e-02) | 77.3557 (±5.5966e-03) | 83.2440 (±6.5455e-03) |
> | oasis | 24.9383 (±1.8569e-02) | 33.5827 (±4.8186e-02) | 27.6380 (±7.0119e-02) | 59.9048 (±4.4011e-02) | 73.0071 (±5.2791e-02) |
> | scannet | 15.2911 (±3.0187e-02) | 24.6885 (±7.0719e-02) | 63.9744 (±4.0399e-02) | 80.2099 (±3.6255e-02) | 85.3210 (±7.3020e-02) |
> | sintel | 35.2239 (±5.1357e-02) | 45.2638 (±9.9847e-02) | 19.8210 (±3.6971e-02) | 42.1712 (±1.5937e-02) | 54.6647 (±1.4493e-02) |

---

> ### Author Response · Authors · 2024-11-25
> **Authors' Kind Reminder to Reviewer kYir**
>
> Dear Reviewer kYir,
>
> Thank you for taking the time to review our paper and provide your valuable feedback. We have submitted detailed responses to your comments and suggestions, addressing the key points raised. **As the discussion is scheduled to end on November 26th, we would greatly appreciate it if you could review our responses and provide any further clarifications at your earliest convenience. Your feedback is extremely important to us.**
>
> Thank you again for your time and effort.
>
> Best,
>
> Authors of Submission 13

---

> ### Author Response · Authors · 2024-11-25
> **Authors' Second Kind Reminder to Reviewer kYir**
>
> Dear Reviewer kYir,
>
> I hope this message finds you well.
>
> Thank you for your time and effort for reviewing our submission. **As the discussion deadline (November 26th) approaches,** **we would greatly appreciate it if you could review our responses and provide any further clarifications at your earliest convenience.** Your feedback is extremely valuable!
>
> Thanks again for your time and effort!
>
> Best,
>
> Authors of Submission 13

---

> > ### Comment · Reviewer_kYir · 2024-11-26
> > **Thanks!**
> >
> > I really appreciate the efforts or additional results and detailed response from the authors.
> > Regarding the Lotus-D, I still feel it is the same as previous discriminative model, especially with a UNet one, and for Lotus-G, it is basically having additional noise channel concatenated as the input. The authors claim the good performance comes from the diffusion priors, but it is kind of vague. And taking a pretrained diffusion model as feature extractor or training from a pretrained diffusion model are not that novel. One possible ablation might be taking the same architecture, and training the model from scratch. Also regarding the Section F of Supp, do we have some comparison with other SoTA on these two tasks?

---

> ### Author Response · Authors · 2024-11-30
> **Author Response to Reviewer kYir**
>
> > One possible ablation might be taking the same architecture, and training the model from scratch.
> >
>
> Thanks for your valuable feedback. We have trained our Lotus-G and Lotus-D models from scratch (without pre-trained diffusion priors), as shown in the table below. All models were trained on the same mixed dataset as in our main paper and evaluated on the NYUv2 and KITTI datasets. **These experiments demonstrates that “the good performance comes from the diffusion priors”, rather than the model structure itself, especially in zero-shot scenarios with limited training data.**  Based on this insight, we would like to emphasize that the goal of our paper is to investigate how to effectively and efficiently leverage diffusion priors for dense prediction tasks.
>
> | Method | NYUv2 AbsRel⬇️ | NYUv2 delta1⬆️ | KITTI AbsRel⬇️ | KITTI delta1⬆️ |
> | --- | --- | --- | --- | --- |
> | Lotus-G | 5.4 | 96.8 | 8.9 | 92.0 |
> | Lotus-G (from scratch) | 17.8 | 75.2 | 32.6 | 48.9 |
> | Lotus-D | 5.1 | 97.1 | 8.1 | 93.1 |
> | Lotus-D (from scratch) | 16.9 | 75.7 | 31.5 | 48.2 |
>
> > And taking a pre-trained diffusion model as feature extractor or training from a pre-trained diffusion model are not that novel.
> >
>
> We appreciate the reviewer’s feedback, but we believe there is a misunderstanding regarding the novelty and contribution of our work.
>
> We agree that adopting pre-trained diffusion models into dense prediction tasks is a promising trend and has shown good performance. **However, existing methods still face challenges in both accuracy and inference time.** This is because they **just directly** adopt pre-trained diffusion models without carefully exploring more suitable formulations for dense prediction tasks. In our paper, **we provide a systematic analysis and conduct extensive experiments** on the key factors of diffusion formulation, **offering significant insights in this area**. Based on these insights (which are not a shot in the dark), we propose a more tailored and efficient adaptation protocol for dense prediction, **rather than simply reusing existing designs or the UNet structure.**
>
> We would like to clarify that contributions to a field do not solely lie in proposing new model structures or very novel paradigms (*e.g.*, single-step diffusion, there are many follow-up works after the famous consistency model, many of them use the similar distillation idea). New insights and a deeper understanding can be obtained when a model or paradigm is **effectively** adapted to new task (*e.g.*, from generation to perception) are equally important and can lead to meaningful advancements. The effectiveness of Lotus is mainly demonstrated by SoTA (or comparable to SoTA) performance with minimal of training data.
>
> We highlight the key contributions of our work:
>
> 1) We systematically analyze the diffusion formulation and find its **parameterization type**, designed for image generation, is unsuitable for dense prediction. Specifically, we observe that $x_0$-prediction has lower variance than $\epsilon$-prediction, leading to improved accuracy.
>
> 2) Based on our analysis, we also find that the **computation-intensive multi-step diffusion process is also unnecessary** and challenging to optimize. We propose fine-tuning the pre-trained diffusion model with fewer training time-steps. Under limited training data, the single-step formulation is the most efficient way to leverage the diffusion priors and achieves promising performance in both accuracy and inference time.
>
> 3) We introduce **a novel regularization strategy,** called Detail Preserver,  to improve prediction accuracy in detailed areas. We also analyze its effectiveness from a frequency perspective in Sec. G of the supplementary material and report the ablation studies in Tab. 3 of the main paper.
>
> > Regarding the Section F of Supp, do we have some comparison with other SoTA on these two tasks?
> >
>
> The comparison of our method with other SoTA foundation models on semantic segmentation is shown in the table below. All other methods are evaluated on the Hypersim dataset for a fair comparison. These experiments demonstrate that the proposed diffusion adaptation protocol, Lotus, has the potential to be applied to other foundational tasks in computer vision. Our model not only outperforms our baseline method “Direct Adaption”, but also achieves comparable performance with SoTA method. We are refining the comparisons on both semantic segmentation and diffuse reflection, *e.g.*, comparing with more SoTA methods, training Lotus with more data, training Lotus with task-specific head, etc.
>
> | Method | mIoU ⬆️ | mAcc ⬆️ |
> | --- | --- | --- |
> | Direct Adaption | 14.1 | 61.3 |
> | DINO(ViT-B) | 19.3 | 62.8 |
> | MAE (ViT-B) | 20.7 | 65.4 |
> | Lotus-G | 21.2 | 65.6 |
> | Multi-MAE (ViT-B) | **22.9** | **66.2** |

---

> > ### Author Response · Authors · 2024-12-01
> > **Authors' Kind Reminder to Reviewer kYir**
> >
> > Dear Reviewer kYir and AC, SAC members,
> >
> > We sincerely thank you for your time and effort for reviewing our submission.
> >
> > **However, as the discussion deadline approaches in less than 3 days, we would greatly appreciate it if you could carefully review our latest responses and provide further clarifications at your earliest convenience. We highly value any opportunities to address any potential remaining issues, which we believe will be helpful for a higher rating. This will be extremely important to us!**
> >
> > Thanks again for your time and effort, we deeply appearciate it!
> >
> > Best,
> > Authors of Submission 13

---

> ### Author Response · Authors · 2024-12-01
> **Authors' Second Kind Reminder to Reviewer kYir (Only 1 Day Left for Reviewers to Post a Message)**
>
> Dear Reviewer kYir,
>
> We sincerely thank you for your time and effort for reviewing our submission, and writing your responses.
>
> **However, as the discussion deadline approaches in less than 2 days (only 1 day left for reviewers to post a message), we would greatly appreciate it if you could carefully review our latest responses and provide further clarifications at your earliest convenience. We highly value any opportunities to address any potential remaining issues, which we believe will be helpful for a higher rating. Your timely feedback is extremely important to us!**
>
> Once again, thanks so much for your time and effort, we deeply appearciate it!
>
> Best,
>
> Authors of Submission 13

---

> ### Author Response · Authors · 2024-12-02
> **Author’s  Additional Response to Reviewer kYir**
>
> We provide additional clarification regarding the novelty of our method.
>
> We uncover a critical yet under-explored principle: we can just fine-tune the pre-trained diffusion model in only single-step with x0-prediction, which is a both more effective and efficient fine-tuning protocol for dense prediction compared to existing methods.
>
> **The exploration process should not be overlooked simply because the final method or model structure is simple. This simple and effective diffusion fine-tuning protocol is obtained through step-by-step systematic analysis, aiming to better leverage the diffusion priors for dense prediction in both accuracy and efficiency.**
>
> Thus, we kindly disagree that our method just simply reuses the UNet structure.
>
> There are some evidences that proves the importance of diffusion priors, rather than the UNet structure itself.
>
> 1. As discussed in this authors’ response (https://openreview.net/forum?id=stK7iOPH9Q&noteId=yjMGqpefHt), the diffusion priors (pre-trained parameters) play a vital role in leveraging diffusion models for dense prediction tasks, where the performance is significantly degraded without diffusion priors, demonstrating that the good performance does not lie in the UNet structure itself.
> 2. The rules of basic diffusion formulations should better be followed. Violating them will lead to performance degradation. For example, in Sec. 4.2 of our main paper, we reduced the number of training time-steps of diffusion formulation to only one, and fixing the only time-step $t$ to $T$ as formulated in Eq. (5), following the basic diffusion formulation (i.e., we denoise from pure Gaussian noise, which corresponds to $t=T$, best matching the diffusion formulation). To further investigate the impact of violating the diffusion formulation, we sampled 5 time-steps at equal intervals—$t=1,250,500,750,1000$—within the range [1,T] (i.e., [1,1000]) and trained 5 diffusion models using these different time-steps. The results are shown in the table below. As observed, the model performs best when $t=T$ ($t=1000$). $t \neq T$ (means the violation of basic diffusion formulation) leads to a slight degradation in performance.
>
> | time-step t | t=1000 | t=750 | t=500 | t=250 | t=1 |
> | --- | --- | --- | --- | --- | --- |
> | AbsRel ⬇️ | **5.587** | 5.631 | 5.727 | 5.663 | 5.737 |
> | delta1 ⬆️ | **96.272** | 96.165 | 96.087 | 96.141 | 96.080 |

---

> > ### Comment · Reviewer_kYir · 2024-12-02
> > **Thanks!**
> >
> > Thanks so much for the follow-up explanations.
> > Could you elaborate the latest results? Are you using pure Gaussian noises with different time steps?
> > My main concerns are,
> > (1) It is not that clear why the noise channels are necessary, given Lotus-D also work well which simply follows the traditional training using paired data and reconstruction loss. The different from previous works are a better / larger pretrained model which was used for training generative models and provided better features. And jointly training images / depth map reconstructions help to preserve the details. But why noises matter? You showed in Fig.9 and demonstrated the uncertainly regions, but could we  expand the discussions?
> > (2) For one-step diffusion, xt (t=1000) contains some very weak information of x0, and it may help the training. Another ablation can be adding pure random noises (break the relationship between x0 and xt) to finetune the pretrained diffusion model. My two-cents are the noise channels hare not important enough to affect the performance, so in this way of training, the performance may not drop too much. It may show that "diffusion" is not necessary for dense prediction.
> >
> > I feel the above explorations are still necessary to better understand the problems of stochastic dense prediction.
> >
> > To be honest, I felt excited when first seeing the title of the paper, since I guessed the paper is about training a vision foundation model from scratch by purely using image data, until I found it a finetuning work with some proposed engineering tricks. And when I read the one-step, I felt excited again and thought it provides some new idea of formulation. So somehow we are leveraging the hot topics to make the paper attractive, but yet not there.
> >
> > However, I really appreciate the detailed rebuttal and the authors' attitudes of research. I believe the paper achieves good performance and can provide some insights for industry. I will increase my score a bit, but encourage the authors to add all these analysis to the Supp. Also do the exploration I mentioned about the random noises.

---

> ### Author Response · Authors · 2024-12-04
> **Author’s Response to Reviewer kYir (Q1-2)**
>
> Dear reviewer kYir,
>
> Thanks so much for your improved rating (5→6)! We are very excited and deeply appreciate it!
>
> Thanks so much for your really thorough and constructive review of our submission and prompt responses! The additional experiments based on your suggestions significantly strengthened the solidness of this submission. We deeply appreciate your valuable comments!
>
> We also sincerely thank you for your further comments to this submission, please see our point-by-point responses to your latest comment below:
>
> > Could you elaborate the latest results? Are you using pure Gaussian noises with different time steps?
> >
> Thanks for your feedback. The noises in our experiments are sampled using the DDPM sampler, as in the original diffusion models, i.e., $x_{t^{\text{noise}}}$=$\sqrt{\bar{a}_{t^\text{noise}}}x_0 $+$ \sqrt{1-\bar{a}_t^\text{noise}} \epsilon $
>
> In our single-step formulation, we fix the time-step $t^{\text{noise}}$ for the DDPM sampler to $t^{\text{noise}}=1000$ . However, to investigate the impact of violating the diffusion formulation,  the time-step $t^{\text{denoiser}}$ for the denoiser model (UNet) can be treated as a hyper-parameter, and we experimented with 5 different values for $t^{\text{denoiser}}$. Our experiments show that the best performance is achieved when the time-step is matched. i.e., $t^{\text{denoiser}}=1000$. We will provide more details about our experiments in our main paper and supplementary material for clarity.
>
> > It is not that clear why the noise channels are necessary, given Lotus-D also work well which simply follows the traditional training using paired data and reconstruction loss. The different from previous works are a better / larger pretrained model which was used for training generative models and provided better features. And jointly training images / depth map reconstructions help to preserve the details. But why noises matter? You showed in Fig.9 and demonstrated the uncertainly regions, but could we expand the discussions?
> >
>
> The main reason we remain the noise input in Lotus-G is that: in our systematic analysis (for better leveraging the valuable diffusion priors for dense prediction, considering both the efficiency and effectiveness), we try to keep the basic diffusion formulation unchanged. A key factor is the stochastic nature.
>
> Strictly speaking, the presence or absence of noise input should belong to two different tasks, because the output of generative version of Lotus (Lotus-G) is a distribution (because the input Gaussian noise represents a two-dimensional Gaussian distribution), and the output of discriminative version of Lotus (Lotus-D) is a single deterministic sample. Considering that dense prediction (and most perception tasks) is originally deterministic, and the evaluation process does not take into account the evaluation of uncertainty, besides Lotus-G, we also proposed Lotus-D to better fit the evaluation settings,  which is a bold attempt. So it is understandable that Lotus-D is slightly better.
>
> Although Lotus-G may be slightly inferior in discriminative benchmarks, does it mean it has no value? Actually not. As we mentioned before, Lotus-G has the advantage over Lotus-D that it retains the input of Gaussian noise, satisfies the basic diffusion formulation, and outputs distribution:
>
> 1. As for dense prediction, although its benchmarks currently hardly support uncertainty evaluation, in fact, for example, in monocular depth estimation, the predictions of the sky; the edge of objects that are parallel with camera rays; detailed objects that are difficult to express clearly under limited pixels such as hair; semi-transparent media (such as water, glasses); fully reflective media (such as mirrors); blurred areas, etc., naturally have uncertainties.
>
> 2. Lotus-D can output one reasonable solution, while Lotus-G can output multiple reasonable solutions for the input image. By analyzing these solutions, such as simply calculating the mean and std (uncertainty) of the depth value of each pixel, then we can infer the uncertainty maps: the area with large std values can be regarded as the area where the model is highly uncertain, i.e., low confidence. For example, when training a self-supervised model, the uncertainty map can assist in excluding low-confidence predictions and only the high-confidence predictions are used for self-training to improve the performance continuously. We believe that the uncertainty map produced by the stochastic model can also support a wide range of interesting downstream applications.

---

> ### Author Response · Authors · 2024-12-04
> **Author’s Response to Reviewer kYir (Q3)**
>
> > For one-step diffusion, xt (t=1000) contains some very weak information of x0, and it may help the training. Another ablation can be adding pure random noises (break the relationship between x0 and xt) to finetune the pretrained diffusion model. My two-cents are the noise channels hare not important enough to affect the performance, so in this way of training, the performance may not drop too much. It may show that "diffusion" is not necessary for dense prediction.
>
> In diffusion models, the forward process progressively adds Gaussian noise to clean samples according to a pre-defined variance schedule, i.e., $\beta_1,\cdots,\beta_T$:
> $q\left(x_t \mid x_{t-1}\right)=\mathcal{N}\left(x_t ; \sqrt{1-\beta_t} x_{t-1}, \beta_t \mathbf{I}\right)$
>
> Let $\alpha_t=1-\beta_t$ and $\bar{a}_t$=$\prod\alpha_s,\ s\in[1,t]$, $x_t$ can be sampled as:
>
> $ q\left(x_t \mid x_0\right)=\mathcal{N}\left(x_t ; \sqrt{\bar{a}_t} x_0,\left(1-\bar{\alpha}_t\right) \mathbf{I}\right)$
>
> Equivalently: $x_t=\sqrt{\bar{a}_t} x_0+\sqrt{1-\bar{a}_t} \epsilon, \epsilon \sim \mathcal{N}(\mathbf{0}, \mathbf{I})$. We can define the SNR as: $\text{SNR}(t)={\overline{a}_t}/{(1-\overline{a}_t)}$. In the standard DDIM scheduler: $\sqrt{\bar{a}_T}\rightarrow0$ and $\sqrt{1-\bar{a}_T} \rightarrow1.0$, which indicates the input, i.e., $x_T$, always contains a small amount of signal of $x_0$ during training.
>
> Breaking the relationship between $x_0$ and $x_t$ to fine-tune the pretrained diffusion model (in our single-timestep setting, it means breaking the $x_0$ and $x_T$), will be a very important ablation. We have implemented the “breaking operation” by forcing the $\sqrt{\bar{a}_T}=0$ and $\sqrt{1-\bar{a}_T} =1.0$, thus, $\text{SNR}(T)={\overline{a}_T}/{(1-\overline{a}_T)}=0$, via setting rescale_betas_zero_snr=True and timestep_spacing="trailing” in schedulers. Also as discussed in [1,2], we believe that removing the leaked GT signal will be very helpful in further understanding the performance of Lotus-G, which preserves the basic diffusion formulation, i.e., the stochastic nature. Due to limited time, our model is still training, we will add these ablation studies in the near future.
>
> We believe that stochasticity (with well-controlled variance in the Lotus-G model, which does not affect accuracy) is a valuable characteristic, and if the performance is comparable with Lotus-D, there seems to be not very necessary to not retain it. Thus, Lotus-G, which adheres to the diffusion formulation, and Lotus-D, which does not strictly follow the diffusion formulation for better fitting the deterministic dense prediction benchmarks, are both proposed in this submission, acting as two different models to meet different needs. Also as we discussed here (https://openreview.net/forum?id=stK7iOPH9Q&noteId=2fqmzk6A1d), when using Lotus-G, as demonstrated in experiments with different time-steps, it is better to strictly follow the diffusion formulation to achieve the best performance.
>
> [1] Fine-Tuning Image-Conditional Diffusion Models is Easier than You Think. ACCV 2024.
>
> [2] Common diffusion noise schedules and sample steps are flawed. WACV 2024.

---

> ### Author Response · Authors · 2024-12-04
> **Author’s Response to Reviewer kYir (Q4)**
>
> > To be honest, I felt excited when first seeing the title of the paper, since I guessed the paper is about training a vision foundation model from scratch by purely using image data, until I found it a finetuning work with some proposed engineering tricks. And when I read the one-step, I felt excited again and thought it provides some new idea of formulation. So somehow we are leveraging the hot topics to make the paper attractive, but yet not there.
>
> Thanks for your feedback! The topics and ideas you mentioned is indeed fascinating! In our paper, due to existing related works suffer from limited performance and slow inference time, our primary goal is to explore how to effectively and efficiently leverage the pre-trained diffusion models for dense prediction. The fine-tuning protocol proposed in this submission is simple but effective and efficient. Moreover, it is applicable to a variety of dense prediction tasks. Now besides depth and normal estimation, experiments in Semantic Segmentation and Diffuse Reflection also support that the proposed fine-tuning protocol works more effectively and effciently compared with directly adapting the pre-trained diffusion model with standard diffusion formulation, as detailed in Sec. F and Fig. D of the supplementary material. For solidness, we will further revise our paper to better highlight our contributions in refining the comparisons on both semantic segmentation and diffuse reflection, e.g., comparing with more SoTA methods, training Lotus with more data, training Lotus with task-specific head, etc.
>
> Overall, thanks again for your prompt response and your kind improved rating! We will update the discussion and corresponding qualitative and quantitative experiments in the camera ready and public archive versions of this submission. Once again, please accept our sincere gratitude for your positive feedback and the improved rating! We sincerely thank you for your time and effort for reviewing our submission, we deeply appreciate it!
>
> Thank you and have a great day,
>
> Authors of submission 13

---

### Author Response · Authors · 2024-11-22
**General Response and Common Concerns**

We sincerely thank all reviewers for their encouraging and insightful comments regarding the novelty and contributions of our paper, such as “clear motivation, in-depth and interesting analysis” (myC3, d5L4, kYir, forA), “plausible and useful detail preserver”(d5L4), “competitive and effective performance”(myC3, d5L4, vkmx) and “clear writing” (forA, vkmx). We will address all reviewer comments point by point and revise our paper accordingly.

Currently, all revisions are written in the supplementary material, here is the added sections:
1. Sec. F. Experiments on More Dense Prediction Tasks: Semantic Segmentation and Diffuse Reflectance
2. Sec. G. Frequency Domain Analysis of the Detail Preserver Take Monocular Depth Estimation as An Example
3. Sec. H. Evaluate the Lotus-G Given Randomness \& Lotus's Performance on DIODE and OASIS
4. Sec. I. Comparison between Lotus-G and Diffusion-E2E-FT (https://arxiv.org/abs/2409.11355, relaesed only about 2 weeks before the ICLR'25 deadline)

We will carefully revise our main paper according to your concerns and our responses.

The following are the common concerns raised by the reviewers.

## Common Concern-1 (for reviewer **kYir, forA, myC3 and vkmx**): More experiments on other tasks

We have additionally trained and evaluated Lotus on two other representative dense prediction tasks, semantic segmentation and diffuse reflection prediction, without any specific modules or loss functions. The training and testing data are obtained in Hypersim dataset.

To validate the effectiveness of Lotus, we compare it with the baseline, Direct Adaption (Fig. 4 in the main paper), as shown in the table below. The results show that Lotus outperforms the baseline across all metrics, demonstrating that Lotus can be effectively applied not only to geometric dense prediction tasks, as shown in the main paper, but also to semantic and rendering dense prediction tasks. We will compare with more recent baseline works in each task, and re-train the Lotus model with similar datasets for more comprehensive experiments. Training Lotus on more tasks is also what we are going to do.

**We have added these results to Tab. A and B, and relative discussions in Sec. F of supplementary material and provided visualization results of Lotus in Fig. D.**

| Semantic Segmentation           | mIoU  ⬆️ | mAcc ⬆️ |
|-------------------|-------|-------|
| Direct Adaption  | 14.1  | 61.3  |
| **Lotus-G**      | **21.2**  | **65.6**  |

| Diffuse Reflection           | L1 Distance⬇️ | L2 Distance (10^-3)⬇️ |
|-------------------|----------------|------------------------|
| Direct Adaption  | 0.198          | 0.206                  |
| **Lotus-G**      | **0.109**      | **0.135**              |

## Common Concern-2 (for reviewer **kYir and vkmx**): Explanations for the detail preserver

The detail preserver is a regularization technique designed to maintain the model's ability to handle detailed areas. When adapting the pre-trained diffusion models to new tasks, the  original ability of diffusion models in processing high-detailed areas  may be undermined, resulting in blurred predictions. As shown in Fig. 8 of our paper, the details like the fences around roads and houses show poor performance. By reconstructing the image using the shared model of dense prediction, the detail preserver, a regularization technique, retains these capabilities.

**We mathematically validate the proposed Detailed Preserver using frequency domain analysis, take depth estimation as an example. We kindly recommend you referring to Sec. G of the supplementary material to see the specific analysis, including visualizations and discussions.**

---

### Meta-Review · Area_Chair_11AW · 2024-12-19

**Metareview:**

The paper introduces Lotus, a new approach to using diffusion models for computer vision tasks like depth estimation and normal prediction. Lotus takes clear images with noise as inputs and produces a dense prediction by predicting annotations instead of noise. The key observations are 1) the standard diffusion model intended for image generation is unsuitable for dense prediction tasks, and 2) the proper fine-tuning protocol for dense prediction tasks can achieve compelling performance in zero-shot depth and normal estimation across various datasets. Technically, the authors analyze prediction types among epsilon or x_0 predictions, and they propose single-step diffusion formulation and detail preserver. Given the compelling results and detailed description of the proposed approach, AC confirms that this paper can shed light on the possibility of the practical use of diffusion priors for dense prediction tasks.

**Additional Comments On Reviewer Discussion:**

Overall, all reviewers recommend the paper's acceptance. There were extensive discussions between reviewers and authors, and AC was impressed by the instant and very detailed feedback from the authors. The main concern raised by reviewers was that more experiments were needed on other tasks, and explanations about 'detail preserver' were required. The author provided additional results regarding semantic segmentation and diffuse reflection predictions, and the authors provided frequency analysis on the detail preserver.

More specifically, reviewer kYir questioned the relationship between diffusion and the UNet-based model, and the authors explain the relationship to conventional task-specific U-shaped networks by comparing it with the proposed Lotus-G or Lotus-D. In addition, the authors addressed Several questions about VAE pre-training, evaluation given randomness, and training from scratch. Reviewre forA raised the question about the novelty of the proposed approach, given that similar attempts have already been made in DDPM, Stable diffusion, or UFOGen. The authors provided feedback regarding their key observations, resulting in compelling results on various dense prediction tasks. The reviewer, myC3, asked questions about the time steps and loss function configuration. The reviewer d5L4 provided extensive comments regarding the performance gap of "direct adaptation" and comparison with "Diffusion-E2E-FT". The reviewer d5L4 also asked about the reason for the larger variability in noise prediction. The authors also provided extensive feedback regarding x0-prediction and AMRN to control the variance. The reviewer d5L4 improved the rating with the request to include the additional contents in the paper. The reviewer vkmx requested more analysis on the time steps, and the authors provided the feedback. The reviewer vkmx was concerned about low mIoU in the Hypersim dataset, but it was posted after the author's feedback deadline. AC noted that this is a rather minor issue that can be addressed in the revision.

---

### Decision · Program_Chairs · 2025-01-22

Accept (Poster)